



# Large-scale sea ice motion from Sentinel-1 and the RADARSAT Constellation Mission

Stephen E.L. Howell[1], Mike Brady[1] and Alexander S. Komarov[2]

[1]Climate Research Division, Environment and Climate Change Canada, Toronto, Canada
[2]Meteorological Research Division, Environment and Climate Change Canada, Ottawa, Canada

*Correspondence to*: Stephen E.L. Howell (Stephen.Howell@ec.gc.ca)

**Abstract.** As Arctic sea ice extent continues to decline, remote sensing observations are becoming even more vital for the monitoring and understanding of sea ice. Recently, the sea ice community has entered a new era of synthetic aperture radar (SAR) satellites operating at C-band with the launch of Sentinel-1A in 2014, Sentinel-1B in 2016 and the RADARSAT
Constellation Mission (RCM) in 2019. These missions represent 5 spaceborne SAR sensors, that together routinely cover the pan-Arctic sea ice domain. Here, we utilized over 60,000 SAR images from Sentinel-1AB (S1) and RCM to generate large-scale sea ice motion (SIM) estimates over the pan-Arctic domain from March to December, 2020. On average, 4.5 million SIM vectors from S1 and RCM were automatically detected per week for 2020 and when combined (S1+RCM) they facilitated the generation of 7-day, 25 km SIM products across the pan-Arctic domain. S1+RCM SIM provided more coverage in Hudson
Bay, Davis Strait, Beaufort Sea, Bering Sea, and over the North Pole compared to SIM from S1 alone. S1+RCM SIM was able to be resolved within the narrow channels and inlets across the pan-Arctic alleviating the main limitation of coarser resolution sensors. S1+RCM SIM provided larger ice speeds with a mean difference (MD) of 1.3 km/day compared to the National Snow and Ice Data Center (NSIDC) SIM product and a MD of 0.76 km/day compared to Ocean and Sea Ice-Satellite Application Facility (OSI-SAF) SIM product. S1+RCM was also able to better resolve SIM in the marginal ice zone compared to the
NSIDC and OSA-SAF SIM products. Overall, our results demonstrate that combining SIM from multiple spaceborne SAR satellites allows for large-scale SIM to be routinely generated across the pan-Arctic domain.

## 1 Introduction

As Arctic sea ice extent continues to decline in concert with increases in carbon dioxide ($CO_2$) emissions (Notz and Stroeve, 2016), remote sensing observations are becoming even more vital for the monitoring and understanding of Arctic sea
ice. Recently, the sea ice community has entered a new era of synthetic aperture radar (SAR) satellites operating at C-band (wavelength, $\lambda$ = 5.5 cm) with the launch of Sentinel-1A in 2014, Sentinel-1B in 2016 (S1; Tores et al., 2012) and the RADARSAT Constellation Mission (RCM) in 2019 (Thompson, 2015). Together these missions represent 5 spaceborne SAR sensors that when combined offer the opportunity to retrieve large-scale sea ice geophysical variables with high spatiotemporal resolution. Small et al. (2021) demonstrated that combining SAR images from S1 and RADARSAT-2 allowed for the





production of high spatiotemporal resolution analysis-ready composite products for large regions. Howell et al. (2019) used analysis-ready composite products generated from S1 and RADARSAT-2 based on the approach described by Small et al., (2021) to provide high spatial resolution estimates of melt onset over a large region in the northern Canadian Arctic.

An important sea ice geophysical variable that could also benefit from large-scale SAR estimates across the Arctic is sea ice motion (SIM). SIM is controlled by the exchange of momentum due to turbulent process primarily from atmospheric

and oceanic forcing. Away from the coast, winds explain 70% or more of the variance in Arctic sea ice motion (Thorndike and Colony, 1982) and as a result, monitoring changes in SIM is important for understanding how sea ice responds to changes in atmospheric circulation (Rigor et al. 2002). SIM convergent and divergent processes impact the overall thickness of Arctic sea ice (Kwok, 2015) and the dynamic component of the Arctic sea ice area and volume balance is also impacted by SIM (Kwok, 2004; Kwok, 2009). The long-term record of SIM in the Arctic indicates the ice speed is increasing which are associated with

thinner ice being more susceptible to wind forcing (Rampal et al., 2007; Kwok et al., 2013; Moore et al., 2019).

Techniques for estimating SIM from satellite observations have a long history dating back to the late 1980s and early 1990s that are primarily based on the maximum cross-correlation coefficient between overlapping images (e.g. Fily and Rothrock, 1987; Kwok et al., 1990; Emery et al., 1991). The maximum cross-correlation approach to estimate SIM can be applied to virtually any overlapping pair of satellite imagery separated by a relatively short time interval of ~1-3 days. For

large-scale SIM, passive microwave imagery is typically the most widely used because of its large swath and daily coverage (e.g. Agnew et al., 1997; Kwok et al., 1998; Lavergne et al., 2010; Tschudi et al., 2020). Enhanced resolution SIM products with spatial resolutions of ~2 km have also been generated (e.g. Haarpaintner, 2006; Agnew et al., 2008) although they have not been widely utilized. The trade-off with respect to large-scale SIM estimated from passive microwave imagery, however, is a low spatial resolution (12-25 km). As a result, SIM is more difficult to track with lower spatial resolution passive

microwave sensors (Kwok et al., 1998) especially, within narrow channels and inlets (e.g. the Canadian Arctic Archipelago; CAA) compared to SAR. However, SIM estimates from SAR are typically regionally based because of image availability across the Arctic.

In this study we make use of 5 SAR satellites from the S1 and RCM missions to generate SIM over the large-scale pan-Arctic domain (Fig. 1). To our knowledge this is perhaps the first time such an extensive combining of SAR imagery at

the pan-Arctic scale has been undertaken to generate SIM. We first describe our workflow that estimates SIM from S1 and RCM SAR imagery (hereafter, S1+RCM) in close to near-real time and combines the output into a S1+RCM SIM product. We then discuss the vector quality and SIM uncertainty of large-scale S1+RCM SIM over the annual cycle for the Arctic and its sub-regions. Finally, we compare our S1+RCM SIM estimates to the existing SIM datasets from the National Snow and Ice Data Center (NSIDC) (Tschudi et al., 2020) and Ocean and Sea Ice-Satellite Application Facility (OSI-SAF) SIM (OSI-SAF)

(Lavergne et al., 2010).



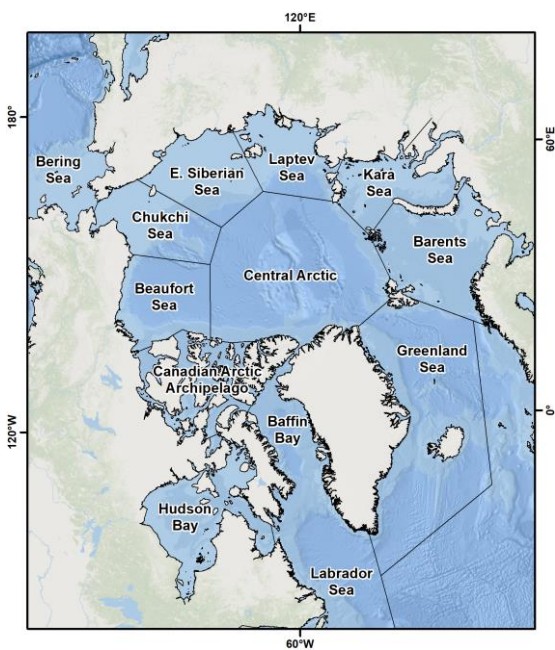

**Figure 1.** Study area domain including sub-regions.

## 2 Data

The primary datasets used in this analysis were Extra Wide Swath imagery at HH polarization from S1 and ScanSAR 50 m (SC50M), ScanSAR 100 m (SC100M), and ScanSAR Low Noise (SCLN) at HH polarization from RCM from March to December 2020 (Table 1). With the recent availability of RCM imagery in 2020, this provided an additional 29,744 images that when combined with S1 (~32,810 images) resulted in 60,000+ images being available to generate SIM across the pan-Arctic for 2020.

We also made use of the 7-day SIM NSIDC Polar Pathfinder dataset and the 2-day OSI-SAF sea ice motion dataset from March to December 2020. Tschudi et al., (2020) provides a complete description of the NSIDC Polar Pathfinder SIM dataset, and Lavergne et al., (2010) provides a complete description of the OSI-SAF SIM dataset. Finally, we used the 2020 daily pan-Arctic ice charts from the National Ice Center.

**Table 1.** Satellite image inventory used in this analysis from March to December 2020.

| Platform | Beam Mode | Pixel Size (m) | Swath (km) | Image Count |
|---|---|---|---|---|
| RCM | ScanSAR 50 m (SC50M) | 20 | 350 | 19,407 |
| | ScanSAR 100 m (SC100M) | 40 | 500 | 9,630 |
| | ScanSAR Low Noise (SCLN) | 40 | 350 | 706 |
| S1 | Extra-Wide Swath (EW) | 40 | 410 | 32,810 |





## 3 Methods

### 3.1 Automated sea ice motion tracking algorithm

We make use of the automated SIM tracking algorithm developed by Komarov and Barber (2014) to estimate large-scale SIM across the pan-Arctic domain. A full description is provided by Komarov and Barber (2014), but the main components of the algorithm are briefly described here. To begin with, coarser spatial resolution levels are generated from the original spatial resolution of the SAR image pairs. For example, if the original SAR image pairs have a spatial resolution of 200 m then the additional generated levels would be 400 m and 800 m. A set of control points (i.e., ice features) is automatically generated for each resolution level based on the SAR image local variances. To highlight edges and heterogeneities at each

resolution level, a Gaussian filter and the Laplace operator are applied sequentially. Beginning with the lowest resolution level, ice feature matches in the image pairs are identified by combining the phase-correlation and cross-correlation matching techniques that allows for both the translation and rotational components of SIM to be identified. SIM vectors not presented in both forward and backwards image registration passes are filtered out, as well as vectors with low cross-correlation coefficients. In order to refine the SIM vectors, at each consecutive resolution level the algorithm is guided by SIM vectors

identified at the previous resolution. An example of the SIM output generated from the algorithm based on two overlapping SAR image is shown in Fig. 2.

The algorithm has been widely utilized for applications that require robust estimates of SIM in Arctic using SAR at C-band (e.g. Howell et al., 2013; Howell et al., 2018; Komarov and Buehner, 2019; Moore et al., 2021). Moreover, when the SIM vector outputs were collocated and coincided with ice buoy trajectories, it was found to have a root-mean square error

(RMSE) of 0.43 km. The limitations that are widely known with respect to estimating SIM from SAR imagery include regions of low ice concentration, melt water on the surface of the sea ice and longer time separation between images also apply to this algorithm.

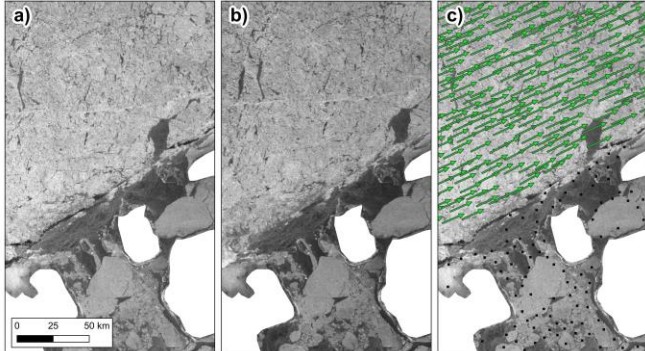

**Figure 2. a) RCM image on April 7, 2020, b) RCM image on April 10, 2020, and c) detected sea ice motion vectors (green) over RCM April 7, 2020. The black dots indicate detection vectors with no motion. RADARSAT Constellation Mission Imagery © Government of Canada 2020. RADARSAT is an official mark of the Canadian Space Agency.**



## 3.2 Generating large-scale gridded sea ice motion

The generalized processing chain for generating large-scale S1+RCM SIM is illustrated in Fig. 3. The approach processes the S1 and RCM image streams separately and then combines the outputs into a S1+RCM SIM product. This parallel approach was chosen because SAR imagery is received by Environment and Climate Change Canada (ECCC) from S1 and RCM in close to near-real time and in order to "keep-up" with the 100's of images coming in per day and the subsequent computational load on automated SIM detection, the processing system is run every hour. As a result, S1 and RCM imagery

are currently not mixed together for automated SIM tracking given the different orbit characteristics of the satellites which contribute to differences in terms of when images are acquired compared to when they are received by ECCC. For example, if an S1 image acquired at 1300h UTC is transferred to our system sooner than an RCM image that was acquired at 1100h UTC then the RCM image would be missed.

    For both S1 and RCM images streams, the pre-processing steps shown in Fig. 3 first involve calibrating the imagery

to the backscatter coefficient of sigma nought ($\sigma°$) using the HH-polarization channel and map-projected to the NSIDC North Pole Stereographic WGS-84, EPSG:3413 coordinate system with a 200 m pixel size. For S1 imagery, pre-processing steps were applied using the Graph Processing Tool (GPT) of the Sentinel Application Platform (SNAP) software, and for RCM imagery, the pre-processing workflow was applied using an in-house pre-processor.

    Manual inspection of SAR imagery and subsequent image stack compilation prior to automatic SIM generation, while

effective in regional-scale studies (e.g. Howell et al., 2013; Howell et al., 2016; Moore et al., 2021) is not practical for generating large-scale SIM. The main challenges of estimating large-scale SIM across the pan-Arctic domain are (i) handling the large volume and delivery frequency of the imagery, (ii) efficiently selecting image stacks, and (iii) providing more computationally efficient feature tracking from the image stacks.

    To address (i) and (ii), an automated approach for determining the suitability of images for inclusion in the automatic

SIM generation (i.e. S1 or RCM image stack selection) was developed and depicted in Fig. 4. For the image stack selection, a 400 km x 400 km grid of sectors encompassing the pan-Arctic was generated and used to create overlapping stacks of SAR image pairs (Fig.4). The footprint geometry of each SAR image was compared to a given sector's extent and if the overlap <= 30% was achieved, that image was retained for feature tracking. Next, we assess image-to-image overlap within each sector to create a temporally sequential image stack that intersected one another to an acceptable degree (>=32,000 km$^2$). Images

within each sector with an overlap of at 32,000 km$^2$ were retained for feature tracking. The stacks are created every hour using the last-processed image from the previous run and the accumulated new imagery that had arrived in the time preceding.

    For (iii), a more computationally efficient application of automatic SIM tracking algorithm (i.e. image stack processing) was developed. Traditionally, image stacks were processed serially which was effective for local-scale studies with limited amounts of imagery, but with significant increases in the SAR image data volume and study area domain size

from S1 and RCM, it was necessary to enhance the processing speed of ice feature tracking analysis. The concurrent approach as outlined in Fig. 5 takes advantage of vertical scalability by increasing the number of processes during image pair analysis. This approach allowed for an entire image stack to be efficiently processed with as many computational cores as were available.





For example, when three sets of image pairs are processing and process 3 finishes before process 2, process 3 picks up the next

sequence of pairs instead of waiting for process 2 (Fig. 5). After stack processing, the last-processed image for the given sector

is recorded in a database and processing ends.

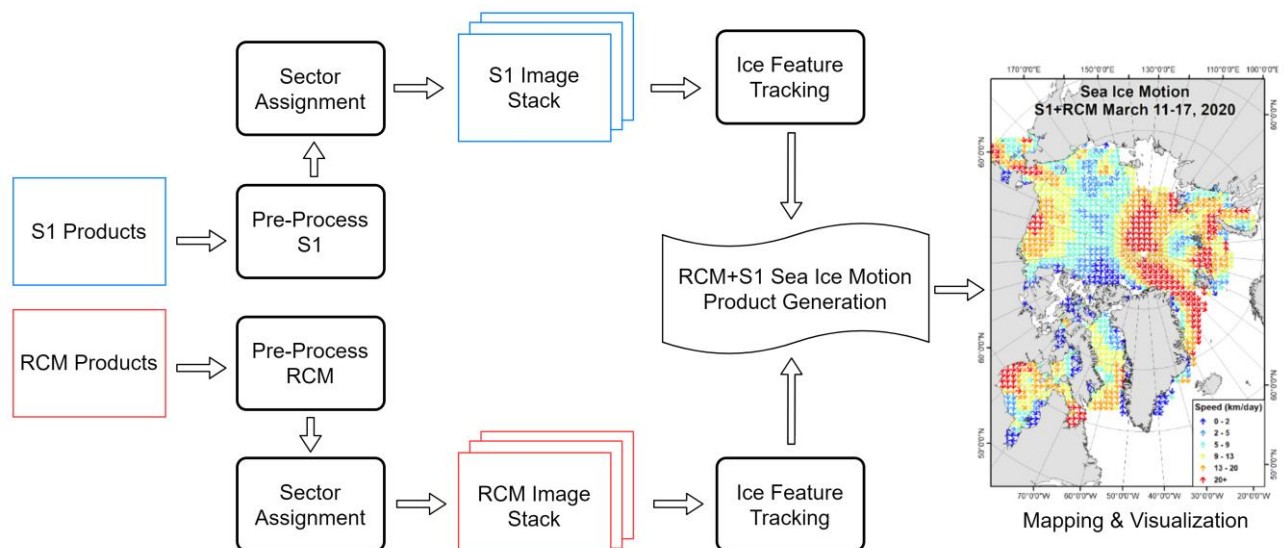

**Figure 3. Generalized processing chain for generating large-scale sea ice motion from S1 and RCM across the pan-Arctic domain.**


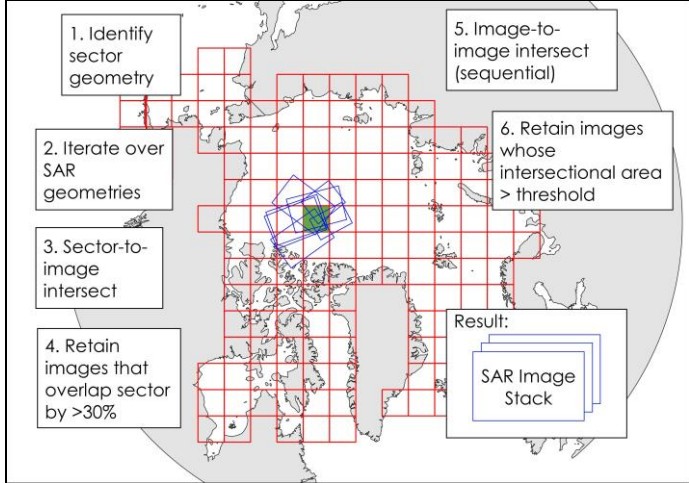

**Figure 4. Processing steps for automatically generating the S1 or RCM synthetic aperture radar (SAR) image stack selection.**





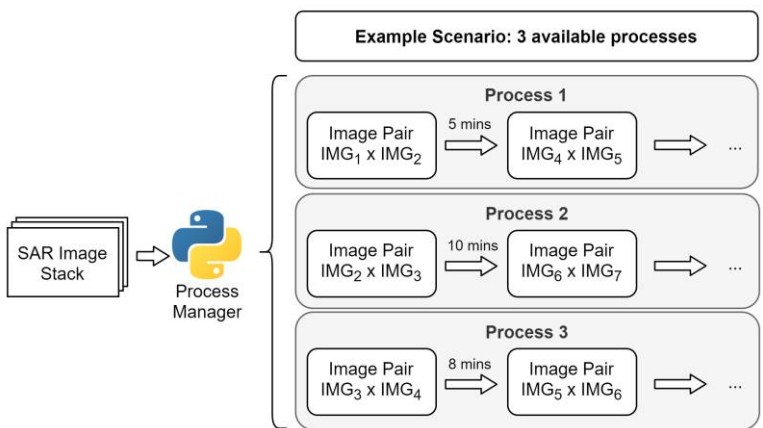

**Figure 5. Illustration of the horizontal scalability approach used to process S1 or RCM image stacks.**

The final step involved combining the results of the automatic S1 and RCM SIM tracking process into defined spatial and temporal resolution grids to be used for analysis and mapping (Fig. 3). Combining the SIM output from S1 with RCM (i.e., S1+RCM) facilitated the ability to improve the spatial coverage and image density of SAR generated SIM across the Arctic. For example, the spatial distribution of SAR image density per week in 2020 for S1, RCM and S1+RCM is shown in Fig. 6. S1 had a denser coverage compared to RCM for the majority of the Arctic regions and especially the Central Arctic

and Greenland Sea. However, RCM coverage was more widely spread across the Arctic compared to S1 and extended into to the Bering Sea, Labrador Sea, and over the North Pole thus filling a gap typically associated with the majority of satellite sensors. An example of the ability of RCM to almost completely cover the North Pole on a single day is shown in Fig. 7. In addition, S1+RCM image density increases with latitude (Fig. 6) indicating that more consistent coverage of the ice pack will be possible during the melt season which is beneficial considering this is when automated SIM tracking algorithms have more

difficulty. Clearly, SAR image density from S1+RCM from March to December 2020 was significant (Fig. 6) with almost complete coverage every 3-days. However, despite the high image density coverage from S1+RCM, more consistent pan-Arctic SIM coverage can be achieved over 7-days because there are more image overlaps during a longer time span. As a result, a 25 km spatial resolution with a temporal resolution of 7-day was selected to provide the most consistent S1+RCM SIM coverage across the pan-Arctic domain. It should be noted that based on Fig. 6 regional S1+RCM SIM products at higher

spatial and temporal resolution are certainly achievable given the image density S1+RCM, and as a result we also briefly demonstrate this capability in the results section.

         For each grid cell, at least 5 individually tracked SIM vectors had to be within a distance of 3 times grid cell resolution cell centroid. Considering the SIM vectors are determined at a spatial resolution of 200 m and gridding takes place at 25 km, numerous vectors are within the grid cell. Only SIM vectors estimated from image pairs with a time separation of greater than

12 hrs were considered.  SIM vectors with speeds greater than 75 km/day where filtered out because based on manual inspection of automatically detected SIM vectors there are sometimes unrealistic anomalous SIM vectors with speeds greater



than 75 km/day. In order to control for SIM speed heterogeneity within a 25 km grid cell, the median SIM was used to represent the ice speed for each grid cell. For each grid cell, a series of descriptive statistics were calculated that included the number of S1 and RCM SIM vectors, the median SIM, the standard deviation of SIM, and the mean cross-correlation coefficient. Even
after removing anomalously large SIM speeds, the automatic SIM tracking algorithm sometimes detected obviously erroneous SIM vectors far from the marginal ice zone and/or near the coast in sufficient quantity (i.e. 5+) to meet the grid cell criteria. These grid cells were subsequently filtered out using a threshold distance of 150 km from the marginal ice zone (i.e., ice concentration of at least 18%) using the weekly National Ice Center ice charts.

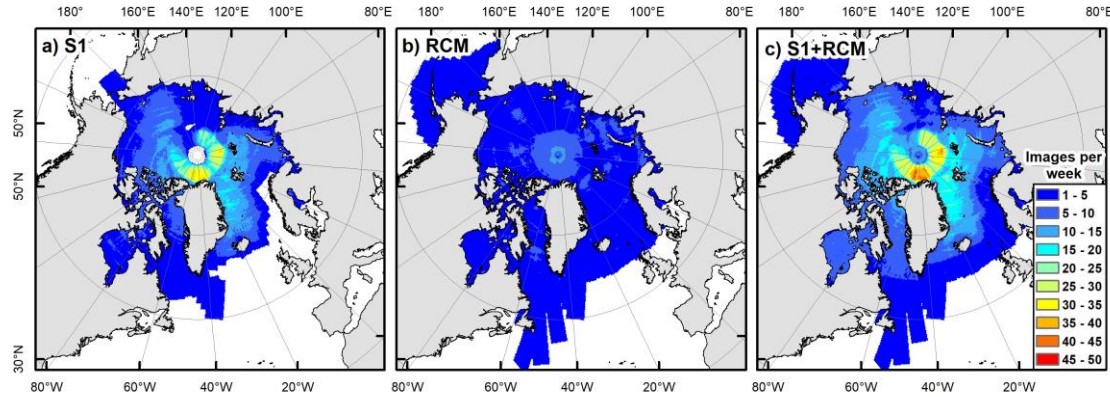

**Figure 6. Image density per week for a) S1, b) RCM, and c) S1+RCM.**

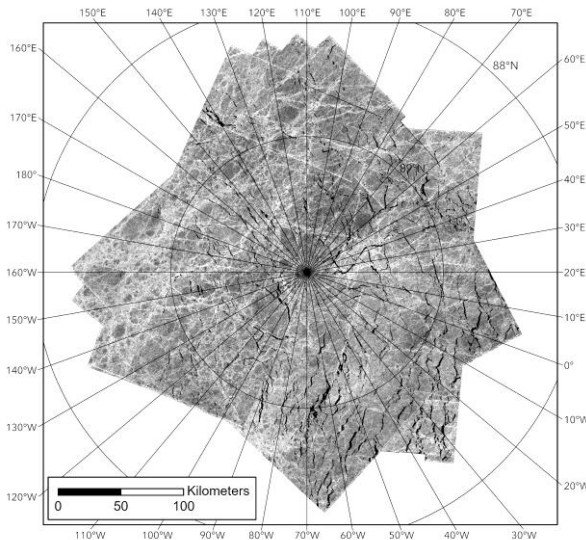

**Figure 7. RCM image coverage over the North Pole on September 15, 2020. RADARSAT Constellation Mission Imagery © Government of Canada 2020. RADARSAT is an official mark of the Canadian Space Agency.**



### 3.3 Quantifying vector quality and uncertainty

The automated SIM tracking algorithm utilized in this study has undergone extensive validation against buoy positions and has an uncertainty of 0.43 km derived for SAR image pairs separated by 1-3 days (Komarov and Barber, 2014). Moreover, SIM output from the tracking algorithm has been found to be in good agreement with other tracking algorithms that includes the RADARSAT Geophysical Processor (e.g. Kwok, 2006; Agnew et al., 2008; Howell et al., 2013). However, considering the application of the tracking algorithm in this study represents considerably larger spatial and temporal domains it is important to assess the quality and uncertainty of the S1+RCM SIM vectors. To provide a quality assessment of the S1+RCM SIM vectors for each grid cell, the cross-correlation coefficient for all S1+RCM vectors in each grid cell were averaged. However, in order estimate the SIM uncertainty of all the S1+RCM vectors in each grid cell, a more structured approach was adopted.

Let us consider a grid cell containing a set of $N$ sea ice velocity vectors $\vec{V}_i$, where $i = 1, 2, \ldots, N$. Each vector has the following uncertainty associated with the SIM tracking algorithm deriving the ice motion vector from two consecutive images:

$$\Delta V_i = \frac{s_0}{\Delta t_i}, \tag{1}$$

where, $\Delta t_i$ is the time interval (in days) separating two SAR images used to derive the considered ice velocity vector $\vec{V}_i$. $s_o = 0.43\,km$ is the uncertainty in sea ice displacement (not speed) reported by Komarov and Barber (2013). Note that $s_o$ was derived for the SAR images separated by a variable time interval (1-3 days), so it must be divided by $\Delta t_i$ to come up with the ice velocity uncertainty. The average velocity value assigned to the considered grid cell is the following:

$$\vec{V}_i = \frac{1}{N} \sum_{i=1}^{N} \vec{V}_i . \tag{2}$$

The SIM uncertainty of each grid cell, $\sigma_{SIM}$ can be estimated as follows:

$$\sigma_{SIM} = \frac{\sigma_0}{\sqrt{1-\alpha^2}}, \tag{3}$$

where, $\sigma_0$ is the base uncertainty given as follows:

$$\sigma_0 = \left\{ \frac{1}{N} \sum_{i=1}^{N} \Delta V_i^2 \right\}^{0.5}, \tag{4}$$

and $\alpha$ is the uncertainty score (varying from 0 to 1) associated with the methodology used to aggregate $N$ individual ice motion vectors derived from pairs of images into the SIM product:

$$\alpha = \left\{ \frac{1}{3} [(c_{max} - \bar{c})^2 + (1 - \tau)^2 + (1 - n)^2] \right\}^{0.5} \tag{5}$$

where, $\bar{c}$ is the average cross-correlation coefficient within each grid cell, $c_{max}$ is the maximum cross-correlation coefficient within each grid cell. $\tau = \frac{t_{SAR}}{T}$ is the fraction of time when SAR imagery are available. Here, $t_{SAR}$ represents the time interval when SAR data were available over the entire time interval considered ($T$). $n = \frac{N}{N_{max}}$ is the relative number of ice motion vectors used to create the aggregated mean ice velocity vector $\vec{V}$. Here, $N$ is the number of ice motion vectors within the cell, and $N_{max}$ is the maximum possible number of the ice motion vectors for a grid cell.



## 4 Results and Discussion

### 4.1 S1+RCM sea ice motion

#### 4.1.1 Pan-Arctic

Table 2 shows the number of SIM vectors detected for the pan-Arctic and each sub-region (Fig. 1) based on over 60,000+ S1 and RCM images available over the period of March to December 2020. On average, there were 4,555,186 SIM vectors detected each week for 2020. The majority (~60%) were located in the Central Arctic sub-region that contains the perennial Arctic sea ice pack and also has a very high weekly S1+RCM image density (Fig. 6). Fig. 8 illustrates the time series of 7-day ice speed from S1+RCM averaged over the entire pan-Arctic domain from March to December, 2020. The ice speed

seasonal cycle as detected by S1+RCM was clear with ice speed decreasing during the melt season and increasing into the fall and winter.

An example of the spatial distribution of S1+RCM SIM on March 11-17, 2020 and December 16-22, 2020 is shown in Fig. 9. Notable features for March include the Transpolar Drift, counter-clockwise SIM characteristic of a Beaufort Gyre reversal, and landfast (no ice motion) ice conditions within the majority of the CAA. The most notable feature for December was the clockwise SIM characteristic of the Beaufort Gyre. For the March 11-17 example, the spatial coverage was extensive

with the exception of a gap within the Laptev Sea which is to be expected based the weekly image density (Fig.6). For the December 16-22 example, the spatial coverage was also extensive and included in the Laptev Sea. Overall, the high density image coverage achieved with S1+RCM was able to provide weekly SAR derived estimates of SIM across the Arctic for 2020.

**Table 2. Average number of S1+RCM sea ice motion vectors detected per week for the Arctic from over the period of March to**
**December 2020.**

| Region | Number of Vectors |
|---|---|
| Baffin Bay/Labrador Sea | 84,163 |
| Barents Sea | 68,522 |
| Beaufort Sea | 534,168 |
| Bering Sea | 11,169 |
| Canadian Arctic Archipelago | 79,316 |
| Chukchi Sea | 369,844 |
| East Siberian Sea | 346,031 |
| Greenland Sea | 136,171 |
| Central Arctic | 2,725,437 |
| Hudson Bay | 89,624 |
| Kara Sea | 104,453 |
| Laptev Sea | 49,771 |
| Pan-Arctic | 4,555,186 |

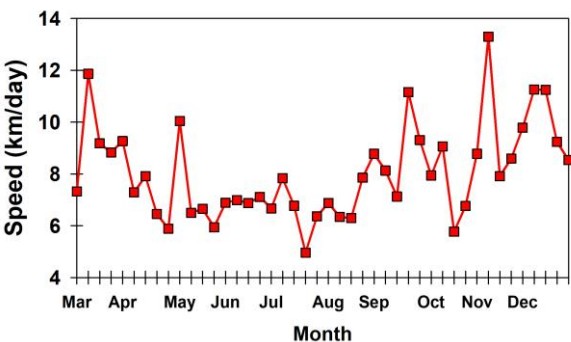

**Figure 8. The time series of 7-day ice speed from S1+RCM averaged over the entire pan-Arctic domain from March to December, 2020. Only ice speeds > 0 were used to calculate the averages (i.e., no zero SIM data were used).**


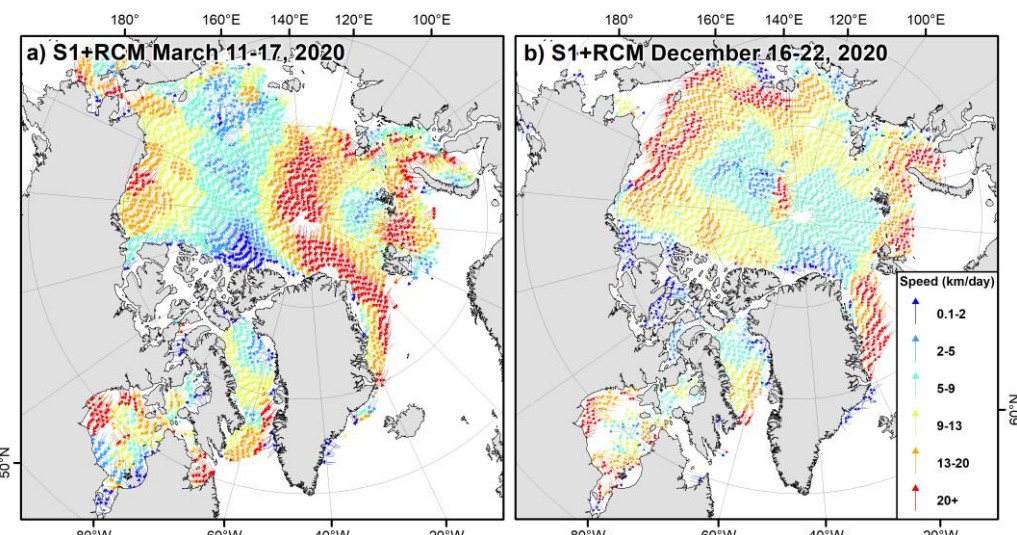

**Figure 9. The spatial distribution of S1+RCM sea ice motion on a) March 11-17, 2020 and b) December 16-22, 2020. Note that the white areas in the figure indicate either zero ice motion for the landfast ice or no ice motion information extracted (because of no SAR data, no ice, or no stable ice features).**


### 4.1.2 Canadian Arctic Archipelago (CAA)

Although not the primary focus of this study, Fig. 6 underscores that in addition to pan-Arctic S1+RCM SIM products, high spatial and temporal resolution regional S1+RCM products are certainly achievable for 2020. To that end, we generated 12.5 km, 7-day S1+RCM SIM for the CAA from March to December, 2020. The CAA is a region where SIM is not typically 270 well resolved from coarser resolution satellites because of its narrow channels and inlets which makes automated SIM tracking difficult, especially during the melt season. Resolving SIM within the CAA (and during the melt season) was achieved because S1+RCM SIM vectors are initially derived at a spatial resolution of 200 m, therefore, alleviating the main limitation of coarser resolution sensors.  The annual cycle of ice speed time series shown in Fig. 10 is representative of the CAA being mostly




landfast from November to July as ice speed is typically slow and confined to the periphery regions (Agnew et al., 2008). The
spike in June was associated with an ice fracture in eastern periphery of the CAA (not shown).  An example of the S1+RCM
SIM spatial distribution from August 12-18 at 12.5 km is shown in Fig. 11 and illustrates the considerable spatial variability
of SIM within the CAA.

Given the considerable weekly image density available from S1 and RCM, further enhances in spatial and temporal
resolution for more local scale SIM studies are also possible (e.g. Moore et al., 2021). Although the focus of this study is
primarily large-scale SIM the latter point is important to demonstrate.  Figure 12 illustrates and example of how SIM can be
resolved from resolutions of 6.25 km, 12.5 km, and 25 km for a 3-day temporal resolution within the middle of CAA. The
level of detail that can be resolved at even 6.25 km is striking and even further increases are possible given the original 200 m
spatial resolution.

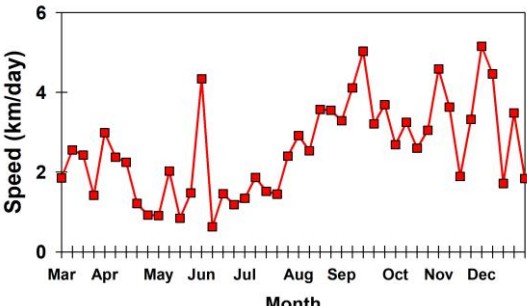

**Figure 10. The time series of 7-day ice speed from S1+RCM averaged over the Canadian Arctic Archipelago from March to December, 2020. Only ice speeds > 0 were used to calculate the averages (i.e., no zero SIM data were used).**

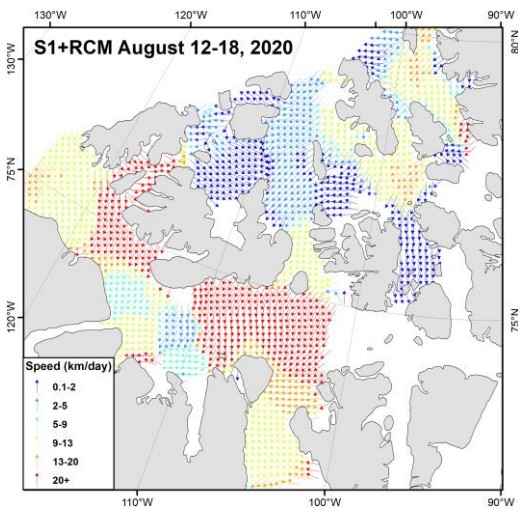

**Figure 11. The spatial distribution of S1+RCM sea ice motion on August 12-18, 2020 in the Canadian Arctic Archipelago. Note that the white areas in the figure indicate either zero ice motion for the landfast ice or no ice motion information extracted (because of no SAR data, no ice, or no stable ice features).**

295

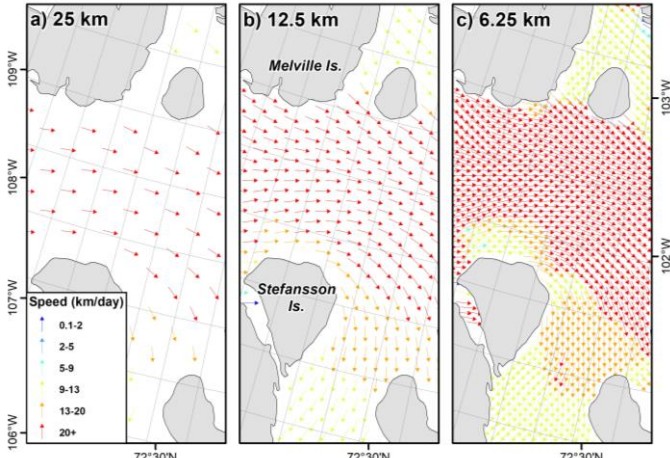

**Figure 12. The spatial distribution of S1+RCM sea ice motion on August 12-18, 2020 in the middle of the Canadian Arctic Archipelago at spatial resolutions of a) 25 km, b) 12.5 km, and c) 6.25 km.**

### 4.2 Spatiotemporal variability of sea ice motion vector quality and uncertainty

Figure 13 shows the time series of the S1+RCM weekly average of the cross-correlation coefficients and $\sigma_{SIM}$ across the pan-Arctic. Note that for automated SIM tracking algorithm used in this study, the cross-correlation coefficients are calculated for the second order derivatives (Laplacians) of the images, and not the original images; therefore, the cross-correlation coefficients may appear lower than reported in the literature by other studies. Both the cross-correlation coefficient and $\sigma_{SIM}$ exhibited the expected variability associated with the seasonal cycle of sea ice and remained relatively high and stable during the dry winter conditions, decreased during the melt season and then returned to stability following the melt season (Fig. 13). $\sigma_{SIM}$ was initially high in early-March because of lower amounts RCM imagery when it first became operational (Fig. 13). As found in previous studies, higher $\sigma_{SIM}$ and lower quality vectors are more apparent during the shoulder seasons (i.e. melt-freeze transitions) as a result of water on the surface of the ice and low ice concentration making automated feature tracking more difficult (e.g., Agnew et al., 2008; Lavergne et al., 2010). Moreover, there are also fewer vectors detected during the shoulder seasons compared to dry winter conditions which contributes to higher $\sigma_{SIM}$.

Figure 14 illustrates the $\sigma_{SIM}$ spatially for selected weekly periods during the 2020 annual cycle. For all cases, $\sigma_{SIM}$ values are typically found in the central Arctic and gradual increases outwards (Fig. 14). The observed spatial variability of $\sigma_{SIM}$ is in part related to weekly SAR image density that decreases away from the central Arctic because they are primarily only covered by RCM (Fig.6). For example, higher $\sigma_{SIM}$ was observed in the periphery regions of the Beaufort Sea, Hudson Bay, and the Bering Sea during March 11-17 (Fig. 14a) and in Lapev Sea for December 16-22 (Fig. 14d). $\sigma_{SIM}$ was lower during the summer months but the weekly image density of S1+RCM (Fig. 6), provided considerably more images over the marginal ice zones during the melt season. As a result, there are more image data to better resolve challenging ice conditions during the shoulder as shown from August 11-17 (Fig. 14b) and September 30-October 7 (Fig. 14c).





Figure 15 summarizes $\sigma_{SIM}$ and the cross-correlation coefficient using box-plots for the each Arctic sub-region from March to December 2020. The interquartile range for most sub-regions were between 0.35-0.45 for the cross-correlation coefficient (Fig. 15a) and between 0.4-0.6 km/day for $\sigma_{SIM}$ (Fig.15b).  The largest $\sigma_{SIM}$ was found in Bering Sea and the lowest in the Central Arctic (Fig. 15b). Lower cross-correlation coefficients were more apparent for regions that contain a significant portion marginal ice zone in 2020 (e.g. East Siberian Sea, Chukchi Sea, and Greenland Sea). Based on Fig.15, the

association between higher (lower)  $\sigma_{SIM}$ and lower (higher) cross-correlation coefficients was not always apparent at the sub-region scale because of the variability of each regions sea ice physical characteristics over the annual cycle. Depending on how ice conditions evolve during the melt season, the distribution of regions with low ice concentrations may vary accordingly. Sub-region  $\sigma_{SIM}$ may also change regionally in subsequent years depending if the weekly pan-Arctic S1+RCM image density changes.  To that end, the regional  $\sigma_{SIM}$ and cross-correlation coefficients values presented in Fig. 15 should be interpreted as

initial or baseline values since we are only considering the year 2020.

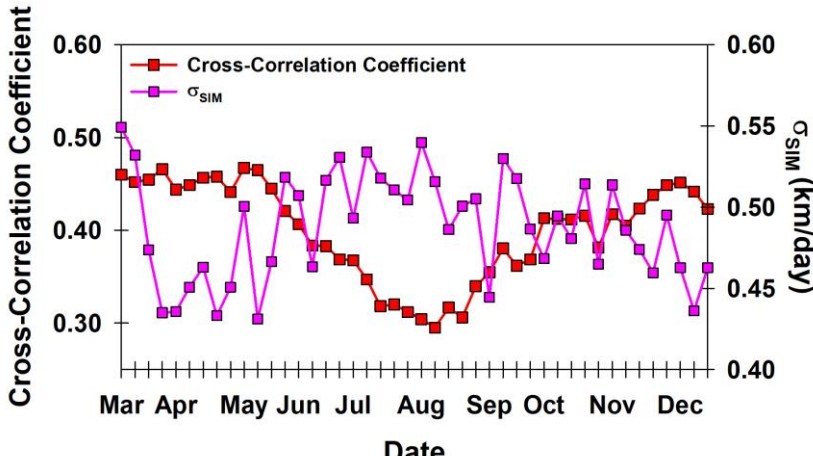

**Figure 13. Time series of the weekly average of sea ice motion cross-correlation coefficient and sea ice motion uncertainty ($\sigma_{sim}$) across the pan-Arctic from March to December, 2020.**


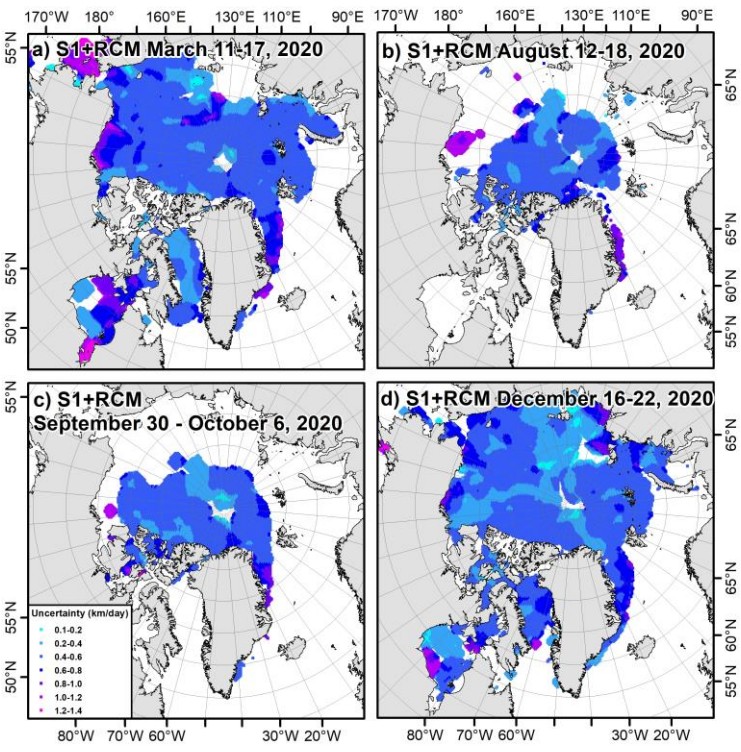

**Figure 14. Spatial distribution of S1+RCM sea ice motion uncertainty on a) March 11-17, 2020, b) August 12-18, c) September 30-October 6, and d) December 16-22, 2020. Note that the white areas in the figure indicate either zero ice motion for the landfast ice or no ice motion information extracted (because of no SAR data, no ice, or no stable ice features).**

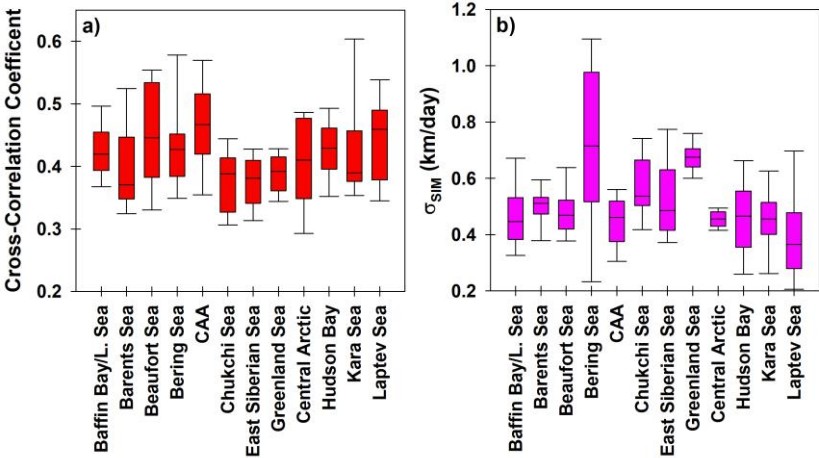

**Figure 15. Boxplots of the S1+RCM a) cross-correlation coefficient and b) sea ice motion uncertainty ($\sigma_{sim}$) for the Arctic sub-regions from March to December, 2020.**





### 4.3 Comparison of S1+RCM against NSIDC and OSI-SAF

Comparing our S1+RCM SIM results with existing NSIDC and OSI-SAF SIM products provides additional quantitative confidence metrics. In order to facilitate a representative 1-to-1 comparison between S1+RCM SIM and both the NSIDC and OSI-SAF SIM products, the spatial and temporal resolution of the S1+RCM were matched with the NSIDC and OSI-SAF SIM products for 2020. For OSI-SAF, S1+RCM was generated with a 2-day at 62.5 km and for NSIDC, S1+RCM was generated with 7-day temporal resolution and 25 km spatial resolution. For each product's temporal resolution (i.e. 7-day

for NSIDC and 2-day for OSI-SAF), all the S1+RCM SIM vectors within each products grid cells (i.e. 25 km for NSIDC and 62.5 km for OSI-SAF) were averaged. The average of all the grid cells within each region across the Arctic (Fig.1) was then determined for all products. This resulted in 343 weekly averages for the S1+RCM and NSIDC comparison and 1957 2-day averages for the S1+RCM and OSI-SAF comparison.  More samples were available from OSI-SAF because of the difference in temporal resolution of these two products (i.e., 2-day vs 7-day)

Scatterplots of S1+RCM versus NSIDC and OSI-SAF are shown in Figs. 16a and 16b, respectively. Both existing SIM products are in good agreement with S1+RCM with correlation coefficients greater than 0.85. Larger speeds are more apparent for S1+RCM with the mean difference (MD) 1.3 km/day for the NSIDC and 0.76 km/day for OSI-SAF for 2020. The root-mean square difference (RMSD) was lower for the NSIDC compared to OSI-SAF at 2.58 km/day and 3.25 km/day, respectively. We note better agreement with NSIDC because of its higher spatial resolution compared to OSI-SAF and the

absence of larger ice speeds in NSIDC compared to OSI-SAF due to lower temporal resolution. The overall larger speed associated with S1+RCM is most likely the result of higher spatial resolution compared to lower resolution satellite data used in NSIDC and OSI-SAF that is more difficult to track at lower spatial resolution because of temporal decorrelation. Kwok et al. (1998) also noted this problem when comparing SIM from passive microwave with SAR and found it also applies to regions of low ice concentration.

Although SAR has difficultly tracking sea ice in the vicinity of the marginal ice zone and regions of low concentration, passive microwave has more difficultly. With the high S1+RCM image density available over the marginal ice zone during the melt season (Fig. 6) there was sufficient number of images available during the summer months for S1+RCM to be able to better resolve SIM compared to these existing SIM products with a lower nominal spatial resolution. For example, the spatial distribution of SIM for S1+RCM and NSIDC product in the Beaufort Sea is shown for September 9-15, 2020 in Fig. 17. While

larger ice speeds are apparent with S1+RCM, there are also many regions along the marginal ice zone with sea ice concentration (SIC) below 18% that are not detected by NSIDC SIM product.





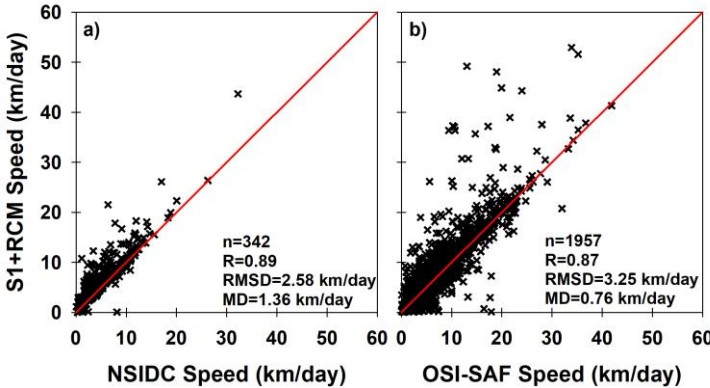

**Figure 16. Scatterplots of S1+RCM sea ice motion versus a) National Snow and Ice Data Center (NSIDC) SIM and b) Ocean and Sea Ice-Satellite Application Facility (OSI-SAF) sea ice motion. Also shown is the number of samples (n), Pearson's correlation coefficient (R), root-mean square difference (RMSD), and the mean difference (MD).**

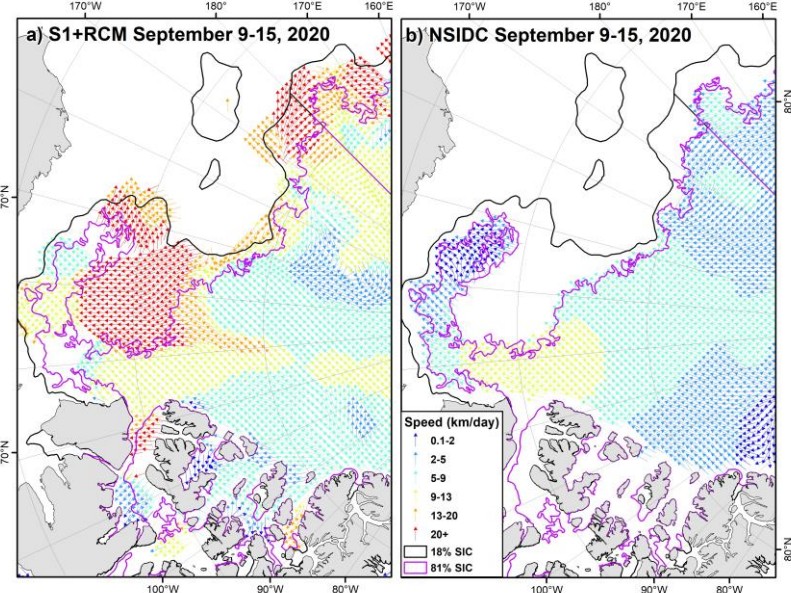

**Figure 17. The spatial distribution of a) S1+RCM sea ice motion and b) National Snow and Ice Data Center (NSIDC) sea ice motion in the Beaufort Sea region from September 9-15, 2020. Also shown are the sea ice concentration (SIC) polygons for 18% (black) and 81% (magenta). Note that the white areas in the figure indicate either zero ice motion for the landfast ice or no ice motion information extracted (because of no SAR data, no ice, or no stable ice features).**

## 5 Conclusions

In this study we made use of 5 SAR satellites from S1 and the RCM with over 60,000 images to estimate SIM over the large-scale pan-Arctic domain from March to December 2020. The higher density image coverage of S1+RCM as oppose to just S1 and/or RCM provided more available SAR image pairs over Hudson Bay, Davis Strait, Beaufort Sea, Bering Sea, and the North Pole. Results indicated that on average, 4.5 million SIM vectors from S1 and RCM were automatically detected



per week for 2020 facilitating the generation of large-scale S1+RCM SIM products. Notable spatial features were apparent (i.e. the Transpolar Drift and the Beaufort Gyre) and the seasonal cycle of sea ice speed exhibited the expected variability with decreases during the melt season and increases into the fall and winter. Moreover, by using an input spatial resolution of 200 m, more spatial heterogeneity in large-scale SIM was preserved as well as SIM was able to be resolved within the narrow channels and inlets across the Arctic. The corresponding S1+RCM SIM vector quality (cross-correlation coefficients) and $\sigma_{SIM}$ also reflected the seasonal cycle with lower quality vectors and higher $\sigma_{SIM}$ during the shoulder seasons. Comparing the S1+RCM SIM estimates to the existing SIM datasets of NSIDC and OSI-SAF revealed that S1+RCM provides larger ice speeds and detects more vectors in the marginal ice zone. The advantages of detecting SIM from SAR as opposed to passive microwave were ultimately confirmed from a large-scale comparison.

S1+RCM SIM covered the majority of the pan-Arctic domain from March to December using a spatial resolution of 25 km and temporal resolution of 7-days.  However, more consistent spatial coverage was achieved during the melt season given the weekly image density of S1+RCM decreases with decreasing latitude. Continued coordinated efforts by working groups like the Polar Space Task Group are encouraged to improve or refine SAR coverage across the pan-Arctic over the annual cycle. Although, covering the majority of pan-Arctic domain with SIM generated from S1+RCM at spatial resolutions of less than 25 km and temporal resolutions less than 7-days was not consistently possible for 2020, we demonstrated that regional S1+RCM SIM products at higher spatial and temporal resolution are achievable given the weekly image density of S1+RCM.

The unique nature of the approach to automatically estimate SIM from S1+RCM described in this paper is that future refinements are possible. For instance, we anticipate adding HV channel to complement SIM estimated from HH polarization. Should the timing of when S1 and RCM images acquired and received by ECCC becomes more consistent, we can explore if mixing S1 and RCM images provides most robust pan-Arctic SIM estimation.  The anticipated launch of the NASA-ISRO (NISAR) L&S-band SAR satellite also provides an opportunity to add L-band into our SIM processing chain. L-band SAR would be able to provide improved SIM estimates during the melt season compared to C-band (Howell et al., 2018). Even without refining our approach, the upcoming launch of Sentinel-1C and Sentinel-1D will continue to facilitate large-scale SIM from C-band SAR for many years to come.

**Data availability**

The S1 imagery is available at the Copernicus Open Access Hub (https://scihub.copernicus.eu/dhus/#/home) and RCM imagery is available online at Natural Resources Canada's Earth Observation Data Management System (https://www.eodms-sgdot.nrcan-rncan.gc.ca). Ice charts from the National Ice Center are available at: https://usicecenter.gov/Products/ArcticData. S1+RCM pan-Arctic SIM products generated in this analysis are available at: https://crd-data-donnees-rdc.ec.gc.ca/CPS/products/PanArctic_SIM/.



**Author contribution**

SELH wrote the manuscript with input from ASK and MB. SELH and MB preformed the analysis.

**Competing interests**

The authors declare that they have no conflict of interest.

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
