# Peer review of "Generating large-scale sea ice motion from Sentinel-1 and the RADARSAT Constellation Mission using the Environment and Climate Change Canada automated sea ice tracking system"

_The Cryosphere, 2021_

## Referee Comment (RC1)

**General comments**

The manuscript demonstrates a weekly sea ice drift product at 25 km resolution derived from SAR acquired by Sentinel-1 and Radarsat Constellation Mission (RCM). It is barely a scientific publication as no new methodology is suggested, no geophysical process is studied and as described below, usefulness of the new product is not well justified.

The idea of combining data from Sentinel-1 and RCM is brilliant, but it is very disappointing to read that 60.000 SAR images with resolution down to 20 meters were used to produce only weekly sea ice drift at only 25 km.

SAR imagery is a treasure for sea ice drift retrieval. Figure 6 clearly shows that in 90% of the Arctic a much more useful sub-daily product could have been generated. Table 2 clearly shows that 2.725.437 weekly vectors in the Central Arctic should not be averaged into a single matrix of 100x100 pixels. And with the input resolution of SAR data a product with spatial resolution of just few kilometers could easily be achieved.

With such high frequency and resolution, the new ice drift product would have become useful for studying highly dynamic processes (fault generation, instantaneous reaction to forcing, inertial oscillations), for detection and tracking of linear kinematic features and evolution of deformation, for validation of ice drift in ice models and for assimilation, for backtracking passive microwave observations for comparison with altimetry, etc. The authors took a very unfortunate decision to blend everything and generate a "consistent" product instead. Yes, the new product is "consistent" and yes, it contains few more ice drift vectors in areas with low concentration. But it doesn't make it more useful than the ice drift product from passive microwave which goes back to 1979. Weekly resolution can be useful for studying sea ice processes at time scales of forty years, but SAR will never achieve that. That's not the purpose of SAR. Obviously, an engineering decision to produce a "consistent" product took over a scientific rationale.

The formulation of uncertainty seems artificial and is not justified either. For example, why would uncertainty increase if the range of cross-correlation coefficients $(c_{max} - c)$ increases? Imagine two cells with all high $c$ in one cell and all low $c$ in another cell. The range $(c_{max} - c)$ is equal in both cells and sigma_SIM is also equal. But on the contrary, since cross-correlation coefficient represent vector quality (Page 18, line 401), I would imagine that the cell with low cross-correlation coefficients have higher uncertainty. The same skepticism can be applied to other components of the score alpha. The expected impact of $t_{SAR}$ on uncertainty is not clear, whereas with more vectors the uncertainty should increase and not the opposite.

Combination of vectors instead of combination imagery from Sentinel-1 and RSN is not well justified either. The explanations regarding timeliness of S1 and RSN data arrival to the datacenter could hold for an operational product. But the presented product is not operational neither by definition (it is a weekly average), nor *de facto* – it is produced for 2020. Combination of SAR images from S1 and RCM into one stack would have at least doubled the number of image pairs for ice drift retrieval and allowed even higher frequency/density of vectors.

In my opinion the manuscript cannot be published in The Cryosphere without thorough analysis of the drawbacks of low resolution and the absence of combination of S1 and RCN imagery. Proper formulation and justification of the uncertainty should also be added.

**Specific comments**

**Section 3.2**

**Lines 114 - 118**

The reasoning for not mixing S1 and RCN imagery is weak and is not applicable here. The presented SIM product is not operational, and its production should not depend on arrival time of data. Ideally the processing chain should be changed to perform only individual preprocessing of S1 and RCN data in parallel branches and do the rest of processing in one stack of images. If that is not feasible, the impact of not combining imagery in a single stack should be clearly presented in the results and discussed. How many pairs of images is produced per week individually from S1 and RCN? How many pairs could have been produced if S1 and RCN data were stacked together? How much the resolution of the end product could have been enhanced? How was that computed?

**Lines 175 – 180**

What is a "consistent" product? This word has many meanings and should be well explained in the current scope. What is the advantage of having "consistent" product? What are the tradeoffs for generating a "consistent" product vs. a useful product? The following statistical analysis is needed for justifying "consistency": relation between decrease in coverage and increase in frequency and resolution. For example, how much can we reduce temporal averaging step and spatial resolution to keep 90% of the Arctic covered by sufficient number of vectors in each cell?

**Line 185**

What is the impact of the threshold of 75 km/day? Where does it come from? Does it mean that vectors with 70 km/day are realistic? How many vectors are rejected? What is the impact of changing the threshold on overall accuracy and number of vectors?

**Line 187**

Why "median" is used for averaging? What is the statistical basis? Is it a normal distribution of ice motion within each cell? Should normalization be applied before averaging, so that mean can be calculated instead of median?

**Section 3.3**

**Line 219 and Lines 229 – 234**

Why is the fixed uncertainty used for both satellites? First, it was shown (Holland et al., 2011; Komarov and Barber, 2014; Korosov and Rampal, 2017) that uncertainty varies from sensor to sensor. Second, the uncertainty is higher for lower cross-correlations.

What are numerical and statistical justifications for the selected formulation of the uncertainty score? Uncertainty is a valuable parameter used, for example, for model evaluation and for assimilation. It should be realistic and reflect the actual spread of drift vectors within a cell. Authors possess vast observations of drift vectors and their RMSE within each cell. And RMSE within a cell is actually a measure of uncertainty. A statistical

analysis of the relationship between the proposed formulation of uncertainty and the actual RMSE should be performed. Such analysis should clearly show impact of each component of the score: *c, tau, n*.

**Section 4.**
**Table 2.**
The purpose of the table is not clear without some extra information. For example, it can help presenting the impact of changed resolution and frequency. The following columns should be present in the table to make it useful:
- Region
- Area
- Number of vectors per week (or per day)
- Number of pixels in the final product containing valid vectors at the following combinations:
  - 7 days, 25 km
  - 3 days, 12.5 km
  - 1 day, 6.25 km

It will hopefully show that even at the highest resolution the number of valid pixels is not dramatically small.

**Figure 9.**
- The vectors are not visible neither in the digital, not in the printed versions of the manuscript. Fewer vectors per inch should be shown and the figures should be rasterized with much higher resolution (at least 300 dpi).
- An example of ice drift in summer and in a shoulder-seasons should be presented. These seasons are especially challenging for ice motion retrieval from SAR.
- A line showing ice edge should be plotted to compare with the extent of the ice drift product.

**Figure 10.**
- Vectors are not visible here either.
- What is the source of patchiness in the drift map? Is it the discrete color scale that enhances gradients? Or is it because various patches were obtained from different image pairs and different sensors? Or is it natural? An explanation and a proof are needed, for example as an extra map showing source of data for each vector by color.

**Section 4.2**
**Line 326**
Due to the ambiguous nature of the alpha score in the uncertainty formulation it is impossible to associate high and low values of sigma_SIM with seasonal variations of sea ice physical characteristics. It could also be due to large range of cross-correlation coefficients, fewer image pairs, uneven distribution of images within a week. Since explicit interpretation of sigma_SIM is impossible the usefulness of the baseline values is not apparent either. As suggested above, the calculated uncertainty should be first related to the observed RMSE of vectors before its values can be interpreted.

**Figure 14.**
The maps with uncertainty look very heterogeneous and for the reasons mentioned above represent rather availability of data than actual uncertainty of the ice drift product. For example, one could expect overall higher uncertainty in summer and shoulder seasons (b and c) than in winter, but the range of values is equal to winter months. The uncertainty formulation should be revised to better reflect the actual spread of drift vectors within a cell.

**Section 4.3**
**Line 356, 357**
What is the purpose of averaging ice drift over a region prior to inter-comparison with OSI-SAF and NSIDC? Thousands of vectors are already averaged in a grid cell, so that spatio-temporal resolution of the tested and the reference products is matching. The intercomparison should therefore be performed on a cell-by-cell basis to better reflect the properties of the product on the resolution as close to the nominal resolution as possible.

**Figure 16.**
In addition to drift speed, comparison of drift direction should also be performed to form a complete picture of difference between the three products. It would also be desirable to plot a map of difference averaged, for example, over seasons.

**Technical corrections**

L20: OSI-SAF

L54: The word "perhaps" seems to need commas around

L54: "… combining of SAR imagery …" can be misinterpreted by a reader with a meaning "Images from S1 were combined with images from RCN to provide SIM vectors". Rephrasing of the sentence is needed which clearly states that drift vectors from S1 and RCN were combined, and not imagery.

L141: What is "vertical scalability"? Some brief explanations (+ a reference) are needed.

L281: "3-day temporal resolution" contradicts the caption on Figure 12: "sea ice motion on August 12 – 18, 2020". Which is correct?

L312: The sentence "For all cases, …" needs to be rephrased. Probably: "For all cases, low sigma_SIM values are typically found in the centra Arctic with gradual increase outwards…"

L324: "significant portion **of** marginal ice zone"

---

## Referee Comment (RC2)

**Review of tc-2021-223 :** *Large-scale sea ice motion from Sentinel-1 and the RADARSAT Constellation Mission* **by Howell and co-authors.**

This manuscript describes a new processing setup for monitoring sea-ice motion at the pan-Arctic scale by taking advantage of satellite imagery from five C-band SAR satellite missions (Copernicus Sentinel-1 A&B, and the three missions from RCM). A first batch (10 months) of S1+RCM sea-ice motion data is prepared and evaluated qualitatively at pan-Arctic scales and regionally in the Canadian Arctic Archipelago. A comparison to two existing large-scale sea-ice motion dataset (from NSIDC and OSI SAF) reveals that the new S1+RCM data generally retrieves faster drift regimes, as well as more vectors in regions with intermediate concentrations and close to land.

The paper provides a description of the processing setup, and conveys well the main message that the recent availability of operational SAR missions opens a new era for large-scale sea-ice motion mapping. The paper is convincing and can be published with some more work.

As I see it, two weak points of the manuscript at this stage are 1) the lack of dedicated quantitative validation of the new S1+RCM drift vectors, namely against trajectories from on-ice drifters, 2) the lack of a stand-alone Discussions section where the choices and assumptions made in the new processing setup and its uncertainties are justified and discussed.

*Major comments:*

**Validation against buoys:**
The paper would be much stronger with a dedicated validation against buoy data at the pan-Arctic scale. Validation against buoy data is the de-facto standard for documenting the accuracy of sea-ice drift datasets (e.g. OSI SAF, NSIDC, Kwok 1998, etc…). In your case it would be particularly useful because validation of RCM SIM vectors (and thus geo-location, resolution, speckle) have never been assessed. You could also check the assumptions built into your uncertainty model (e.g. the scaling of the uncertainty on velocities by Delta_t, see discussion below). I strongly suggest that a dedicated validation against buoy data is conducted and reported here, but leave it to the Editor to decide if this major revision is required or not.

**Sea-ice motion technique:**
Section 3.2 is missing some details to fully characterize the processing. Some of the missing elements are:
* how old are the scenes allowed to be before they are not taken in the stack of scenes?
* In Fig. 5: we see that IMG1xIMG2, IMG2xIMG3, IMG3xIMG4, etc… are processed for SIM, but what about IMG1xIMG3, IMG1xIMG4, etc…? Considering these overlaps would dramatically increase the number of retrieved vectors and the sampling in the temporal domain. Please indicate if these additional overlaps are processed for SIM and, if not, add a discussion/justification why they were not considered (e.g. in a Discussions section).
* In Fig. 3: it is clear and well justified that S1 and RCM scenes are processed on their own (before the merging step). Are SIM vectors processed within the S1 and RCM missions? E.g. S1a with S1b, RCMa with RCMc, etc… Please add this information.
* Fig 6 a) gives the impression that S1 has a complete coverage of the dark blue region at least once on every week. Is it really the case, or are there weeks were S1 leaves some holes in the weekly coverage? Could these [0-1] average density be in a different color to better appreciate the weekly coverage? Same for b).
* L184: what is the justification for the cap at minimum 12 hours?
* starting L183: it is not immediately clear that you average the velocity vectors instead of the displacement vectors. Please clarify.

**Uncertainty parametrization:**
Section 3.3 presents an approach to uncertainty characterization. To date, there are no established procedures or algorithms for attributing per-pixel uncertainties to sea-ice motion vectors, yet alone for merged/averaged vectors like the S1+RCM ones. It is commendable that the authors design such an uncertainty model, but a discussions of the choices made would strengthen the paper.

In the absence of dedicated validation against buoys, the authors chose to base their uncertainty estimates on earlier results by Komarov and Barber (2014). However, these were obtained for a limited number of RADARSAT-2 scenes. Please discuss why the RADARSAT-2 results can be extended to RCM and S1.

L220: is it really the case that the uncertainty of the velocity is the uncertainty of sea-ice displacement divided by the Delta_t ? This should be better justified. For example take Fig. 5 in Lavergne et al. (2021): the RMSE of displacements increases slightly with time-separation Delta_t but not at a rate where twice the time-separation, twice the RMSE in displacement. The RMSE in displacement is controlled by the image "sharpness" and the presence of trackable features. Longer time separation can lead to larger decorrelation, but it is not intuitive that the uncertainty of a drift velocity from a pair of images with Delta_t = 12h is half that from a pair of images with Delta_r = 24h. If you keep this approach, please discuss and justify it more in the text.

The scaling of individual uncertainties into the merged uncertainties by Eq. 5 has some merit. Consider explaining in the text why such a parametrisation was chosen: what are the expected outcome in case, e.g. only a small temporal fraction of the week was covered by SAR scenes, etc… This will help the reader in addition to the description of the terms.

The value of the MCC on its own is not necessarily a good indicator of the vector quality (e.g. Hollands et al. 2015). Intuitively, a sub-window with no features (uniform grey) will match perfectly (MCC=1) with another grey sub-window, yet there will be no peak. Please discuss if the value of the MCC is a good metric of quality.

T. Hollands, S. Linow and W. Dierking, "Reliability Measures for Sea Ice Motion Retrieval From Synthetic Aperture Radar Images," in *IEEE Journal of Selected Topics in Applied Earth Observations and Remote Sensing*, vol. 8, no. 1, pp. 67-75, Jan. 2015, doi: 10.1109/JSTARS.2014.2340572.

**Comparison to NSIDC and OSI SAF:**
Fig. 16 you compare the products in terms of absolute velocities, while the products deliver velocities along x and y components. In Lavergne et al. (2010) we note that: "The transformation from vector components to total velocities is strongly non linear and is known to create artificial pseudobiases that might hinder valuable conclusions to be drawn [Stoffelen, 1998, Appendix B]". Did you try to compare the products in terms of x/y components and did you find similar results?

Stoffelen, A. (1998), Towards the true near-surface wind speed: Error modeling and calibration using triple collocation, *J. Geophys. Res.*, **103**, 7755–7766.

L364-365: the 25km resolution of the NSIDC product is mainly oversampling (the original PMR input data have a coarser resolution, see Table 2 of the NSICD V004 User Guide).

**Other:**
L243-245: it is not intuitive to me that such a seasonal cycle as shown Fig 8 is expected. During summer melt, the ice is thinner and more free to move when pushed by winds (smaller internal stress). In fact Rampal et al. (2009) Fig 4.c find a different seasonal cycle from IABP buoys, with a ramp-up during summer and a maximum in September. Please discuss.

*Minor comments:*

L49: The dataset based on passive microwave indeed have coarse resolution, that rather are in the range (50 – 100 km) than (12-25 km) as stated here. The OSI SAF is ~60km like the data from IFREMER/CERSAT, Kwok's is ~100km. NSIDC's 25km grid results from oversampling (see e.g. Table 2 of the NSIDC V004 User Guide).

L75: Did you use the multi-sensor OSI SAF product (multi-oi) or the single-sensor products (from AMSR2, SSMIS, etc…) Please provide this information.

Fig 14: the labels and legends are hardly readable. Please enlarge the text.

Conclusions: with "swath-to-swath" approach, SIM from passive microwave now achieves sub-daily temporal resolution (Lavergne et al. 2021). This will be extremely difficult to reach consistently and pan-Arctic from SAR constellations alone. Maybe the complementary of SIM estimation from SAR and "swath-to-swath" PMW would deserve a mention in the Conclusions.

*Editorials:*

L15: delete "able to be"

L18-19: OSI SAF, without "-" (in long form and acronym).

L49: "trade-off with respect to" → "drawback of" or "limitation of".

L73: replace "2020" with "this period".

L86: here and later in the section: "coarser spatial resolution levels". Consider changing "levels" with "images" for clarity.

L90. Lowest resolution → coarsest resolution

L135. "at _least_ 32,000 km2"

L250. Based _on_ the weekly image….

---

## Author Response (AR1)

**Reviewer #1**
General comments
The manuscript demonstrates a weekly sea ice drift product at 25 km resolution derived from SAR acquired by Sentinel-1 and Radarsat Constellation Mission (RCM). It is barely a scientific publication as no new methodology is suggested, no geophysical process is studied and as described below, usefulness of the new product is not well justified.

The idea of combining data from Sentinel-1 and RCM is brilliant, but it is very disappointing to read that 60.000 SAR images with resolution down to 20 meters were used to produce only weekly sea ice drift at only 25 km.

**Howell et al.**
The Reviewer makes some good points which we have addressed. However, the Reviewer has provided comments for their opinion of what the manuscript (and inherently SAR SIM) should be which is very subjective and, therefore, there are many other points we disagree with for which we re-butt accordingly.

***Summary of the Major Changes:***
1. We recast the manuscript to describe the Environment and Climate Change Canada Automated Sea Ice Tracking System (ECCC-ASITS) which is to provide routine SIM products from S1+RCM for operational needs at ECCC, the broader scientific community, and maritime stakeholders. The focus is on the latter. Accordingly, our choices, assumptions, and uncertainties are better justified throughout the manuscript
2. Generated a new pan-Arctic 6.25 km 3-day (rolling) S1+RCM SIM product and all datasets have been updated until October 31, 2021.
3. Added validation section that compares vector displacement from S1 and RCM to buoys from the IABP
4. Refined the uncertainty of the S1+RCM SIM products based on the buoy analysis and the time separation of the image pairs for dry and wet ice conditions
5. Provided a 1-to-1 grid cell comparison of NSIDC and OSI-SAF SIM products to S1+RCM

**Reviewer #1**
SAR imagery is a treasure for sea ice drift retrieval. Figure 6 clearly shows that in 90% of the Arctic a much more useful sub-daily product could have been generated. Table 2 clearly shows that 2.725.437 weekly vectors in the Central Arctic should not be averaged into a single matrix of 100x100 pixels. And with the input resolution of SAR data a product with spatial resolution of just few kilometers could easily be achieved.

**Howell et al.**
A sub-daily product is very spotty even in the Central Arctic, and that data would only useful for a targeted study (high temporal resolution) which is not what our study is focused on. There are a lot of redundant vectors despite the large number shown in Table 2, and a sub-daily product cannot be generated as easily as this Reviewer would seem to suggest. We have changed the legend of the imagery density Figure 3 (see below) to reflect this and it points out that some regions are very indeed dense and some are not. We also note that even a daily SAR SIM time

series is challenging in the vicinity of Nares Strait (e.g. see Moore et al., 2021a). Just because a lot of imagery is available does imply automatic vector detection will be successful at the sub-daily scale.

[Figure]

**Figure 3.** Image density per week for a) S1, b) RCM, and c) S1+RCM based on images from March 2020 to October 2021.

However, we do agree we could have pushed the pan-Arctic S1+RCM products more and we have added another product in the processing chain that produces 3-day rolling S1+RCM SIM at 6.25 km (rolling). Examples are shown below but we maintain that given the image density of S1+RCM 3-day is optimal and 7-day is more complete as follows:

[Figure]

**Figure 9.** The spatial distribution of S1+RCM sea ice motion at 25 km on a) July 1-7, 2020, b) August 5-11, 2020, c) July 7-13, 2021, and d) August 4-10, 2021. Note that the white areas in the figure indicate either zero ice motion for the landfast ice or no ice motion information extracted (because of no SAR data, no ice, or no stable ice features).

[Figure]

**Figure 10.** The spatial distribution of S1+RCM sea ice motion on March 12-14, 2020. The letters correspond to zoomed in regions on the map. Note that the white areas in the figure indicate either zero ice motion for the landfast ice or no ice motion information extracted (because of no SAR data, no ice, or no stable ice features).

**Reviewer #1**
With such high frequency and resolution, the new ice drift product would have become useful for studying highly dynamic processes (fault generation, instantaneous reaction to forcing, inertial oscillations), for detection and tracking of linear kinematic features and evolution of deformation, for validation of ice drift in ice models and for assimilation, for backtracking passive microwave observations for comparison with altimetry, etc. The authors took a very unfortunate decision to blend everything and generate a "consistent" product instead. Yes, the new product is "consistent" and yes, it contains few more ice drift vectors in areas with low concentration. But it doesn't make it more useful than the ice drift product from passive microwave which goes back to 1979. Weekly resolution can be useful for studying sea ice processes at time scales of forty years, but SAR will never achieve that.
That's not the purpose of SAR. Obviously, an engineering decision to produce a "consistent" product took over a scientific rationale.

**Howell et al.**
This is a subjective comment. The aim of our study is not on highly dynamic sea ice processes as the Reviewer thinks we should have done. We further disagree with this Reviewer's opinion of the purpose of SAR for SIM studies. For instance, there are other communities and maritime stakeholders that can benefit from newer products and with the availability of SAR imagery from S1 and RCM and new large-scale SIM products based on higher quality datasets are important to produce. These non-scientific interests include (but not limited to): resource extraction and development of infrastructure, fisheries, expedition cruise ships, Indigenous peoples, icebreakers, explorers crossing the ice, ice camp logistics, and search and rescue (SaR). Moreover, in the Canadian Arctic Archipelago S1+RCM SIM fills a major gap because SIM is not resolved by current passive microwave datasets, which has implications for both stakeholder and scientific interests. Finally, even though our dataset time period begins in March 2020, the Arctic is rapidly

changing, and S1+RCM can find utility for recent process studies, so time series creation must be started sooner than later given the current and future availability of SAR.

We articulate this in the Introduction as follows:
With the availability of SAR imagery from S1 and RCM, a new opportunity exists to provide both the operational and scientific communities with larger-scale estimates of SIM from SAR. In addition, with marine activity in the Arctic increasing (e.g. Eguíluz et al., 2016, Dawson et al., 2018), a wide-range of maritime stakeholders could benefit from access to large-scale SAR SIM for safety, planning and situational awareness (Wagner et al., 2020).

**Reviewer #1**
The formulation of uncertainty seems artificial and is not justified either. For example, why would uncertainty increase if the range of cross-correlation coefficients (cmax – c) increases? Imagine two cells with all high c in one cell and all low c in another cell. The range  (cmax – c) is equal in both cells and sigmaSIM is also equal. But on the contrary, since crosscorrelation coefficient represent vector quality (Page 18, line 401), I would imagine that the cell with low cross-correlation coefficients have higher uncertainty. The same skepticism can be applied to other components of the score alpha. The expected impact of tSAR on uncertainty is not clear, whereas with more vectors the uncertainty should increase and notthe opposite.

**Howell et al.**
The Reviewer makes a fair point about uncertainty. We have chosen to revise this approach and just present the average cross-correlation for the grid cell together with a separate formulation of uncertainty based on comparison with buoys. Therefore, the average cross-correlation coefficient, number of vectors, and image coverage for each grid cell are all provided as part of the product, so a user could further assess the quality of the SIM vector in a given grid cell. The revised the section as follows:

Based on the comparison we developed two uncertainty estimates for dry and wet sea ice conditions as follows:
    In order to estimate the SIM uncertainty from the ECCC's automated SIM tracking algorithm, we compared SIM displacement vectors from S1 and RCM to buoy positions from IABP during winter and summer time periods.  For all S1 and RCM displacement vectors (derived from image pairs), the closest buoy trajectory was co-located to the start of each displacement vector position. The distance between the starting point of a given SAR ice motion tracking vector and the starting point of the corresponding buoy trajectory did not exceed 3 km. Fig. 14 summarizes the comparison results for dry winter conditions (April 2020 and 2021) and during the melt season (August 2020 and 2021). The ECCC automated SIM tracking algorithm performs very well during winter conditions with a root mean square error (RMSE) of 2.78 km and a mean difference (MD) of 0.40 km.  The RMSE is higher than the value reported by Komarov and Barber (2014) likely because more image pairs over a larger geographical area were used in this comparison as well as the spatial resolution was lower. Performance slightly decreases during the summer with a lower number of vectors detected and an RMSE of 3.43 km.

[Figure]

**Figure 14.** Comparison between ice motion vectors derived by the Komarov and Barber (2014) automated sea ice tracking algorithm from S1 and RCM SAR images and buoy data.

Taking into consideration the difference between the winter and the summer we assign two uncertainties to the S1+RCM SIM products for dry and wet conditions as follows. Consider a grid cell containing a set of $N$ sea ice velocity vectors $\vec{V}_i$, where $i = 1,2, \dots, N$. Ice speed for this each vector has the following uncertainty associated with the SIM tracking algorithm deriving the ice motion vector from two consecutive images:

$$\Delta V_i = \frac{s_0}{\Delta t_i},$$ (1)

where $\Delta t_i$ is the time interval (in days) separating two SAR images used to derive the considered ice velocity vector $\vec{V}_i$. In (1) $s_o$ is the uncertainty in sea ice displacement (not speed) for dry ice conditions (2.78 km) or wet ice conditions (3.43 km). Note that $s_o$ must be divided by $\Delta t_i$ to come up with the ice velocity uncertainty. The average uncertainty for dry ($s_o = 2.78$ km) and wet ($s_o = 3.43$ km) ice conditions in each grid cell (N) is then determined using the following equation:

$$\sigma_{SIM} = \frac{1}{N} \sum_{i=1}^{N} \Delta V_i$$ (2)

[Figure]

**Figure 15.** Spatial distribution of (a) dry and (b) wet S1+RCM SIM uncertainty for August 5-11, 2020

Fig. 15 shows some an example of the spatial distribution of both dry and wet uncertainty estimates indicating higher estimates for the later. We acknowledge that it is difficult to quantify the impact of SAR image pair availability over 7-days together with automatic SIM vector detection under certain environmental conditions. The number of S1+RCM SIM vectors used in the grid cell generation can subsequently be used to account for this whereby, more confidence (less uncertainty) in SIM can be associated with a larger number of vectors. Moreover, S1+RCM image density increases with latitude (Fig. 3) indicating that more consistent coverage is available over the Central Arctic which, is also beneficial during the melt season when automated SIM tracking algorithms have more difficulty. However, SAR image pair coverage could be exceptional over the 7-day time window, yet environment conditions (e.g. melt ponds, low ice concentration, marginal ice zone, etc.) could still make automatic SIM vector detection difficult resulting in a low number of SIM vectors in the grid cell. The problem of image coverage is less of a concern for the 3-day product given the average image separation is ~2-days. Given the difficultly in quantifying SAR image pair coverage on S1+RCM SIM uncertainty, we now compare S1+RCM SIM to existing products with different temporal resolutions that provides additional quantitative confidence metrics to assess the quality of the S1+RCM SIM estimates.

**Reviewer #1**

Combination of vectors instead of combination imagery from Sentinel-1 and RSN is not well justified either. The explanations regarding timeliness of S1 and RSN data arrival to the datacenter could hold for an operational product. But the presented product is not operational neither by definition (it is a weekly average), nor de facto – it is produced for 2020. Combination of SAR images from S1 and RCM into one stack would have at least doubled the number of image pairs for ice drift retrieval and allowed even higher frequency/density of vectors.

In my opinion the manuscript cannot be published in The Cryosphere without thorough analysis of the drawbacks of low resolution and the absence of combination of S1 and RCN imagery. Proper formulation and justification of the uncertainty should also be added.

**Howell et al.**
Our initial submission was not cast appropriately for what it is and this seems to have resulted in this Reviewer's comments being targeted to an analysis of a smaller image regions with high temporal resolution. Accordingly, mixing S1 and RCM are just not applicable in the system that routinely processes 100's of SAR images per day nor will it significantly improve spatial coverage. It may improve temporal resolution, but small scale processes are not the focus of this study. In addition, the product is not operational, but the system feeds operations and in order to do that routinely (efficiently) the streams have to be considered separately. A rolling 7-day product is sent the Canadian Ice Service (CIS) daily to help the ice forecasters with ice chart analysis. A weekly average product, generated routinely is in fact useful for operational ice forecasters in constructing weekly ice charts. These items are now clearly presented in the manuscript as follows:

The ECCC-ASITS facilitates the routine generation of S1+RCM SIM products; however, it should be noted the system has roots (i.e. built-up) in previous studies (e.g. Howell and Brady, 2019; Moore et al., 2021a; Moore et al., 2021b). While the primary system methodology described here is for larger-scale SIM generation, it is not strictly limited for this application and can (has) been modified to accommodate specific research or operational objectives.

The generalized processing chain for generating large-scale S1+RCM SIM using ECCC-ASTIS is illustrated in Fig. 2. The approach processes the S1 and RCM image streams separately and then combines the outputs into a S1+RCM SIM product. This parallel approach was chosen for several reasons. First, mixing S1 and RCM primarily improves spatial coverage as RCM mainly fills in the spatial gaps in S1 coverage as RCM coverage is more widely spread across the Arctic and covering Bering Sea, Laptev Sea, Davis Strait, Southern Beaufort Sea and even the North Pole thus filling a gap typically associated with the majority of satellite sensors (Fig. 3). An example of the ability of RCM to almost completely cover the North Pole on a single day is shown in Fig. 4. However, we note that the temporal resolution of SIM could be improved by mixing S1 and RCM, but this would be restricted to only certain regions of the Arctic. Second, SAR imagery is received by ECCC from S1 and RCM in close to near-real time and in order to "keep-up" with the 100's of images coming in per day and routinely generate products every day, the processing system is run every hour given computational load on automated SIM detection. Fig 5. Illustrates the amount of S1 and RCM SAR imagery that was processed over a 7-day time period in March 2020 which amounted to over 1132 SAR images or ~160 images per day. Finally, the different orbit characteristics of the satellites which contribute to differences in terms of when images are acquired compared to when they are received by ECCC. For example, if an S1 image acquired at 1300h UTC is transferred to our system sooner than an RCM image that was acquired at 1100h UTC then the RCM image would be missed. We note that S1A and S1B are freely mixed in the Sentinel processing chain as well as RCM1, RCM2 and RCM3 are mixed in the RCM processing chain.

[Figure]

**Figure 5.** Spatial distribution of S1 and RCM SAR images from March 11-17, 2020.

**Reviewer #1**
Specific comments
Section 3.2
Lines 114 - 118
The reasoning for not mixing S1 and RCN imagery is weak and is not applicable here. The presented SIM product is not operational, and its production should not depend on arrival time of data. Ideally the processing chain should be changed to perform only individual preprocessing of S1 and RCN data in parallel branches and do the rest of processing in one stack of images. If that is not feasible, the impact of not combining imagery in a single stack should be clearly presented in the results and discussed. How many pairs of images is produced per week individually from S1 and RCN? How many pairs could have been produced if S1 and RCN data were stacked together? How much the resolution of the end product could have been enhanced? How was that computed?

**Howell et al.**
We would agree with the broad strokes in Reviewer #1's hypothetical scenario but it is not currently possible to implement this workflow in an automated fashion with the available computing infrastructure. As the Reviewer notes, this is not yet an operational product but it does feeds into operations at ECCC and thus we continue to identify pathways to best-address the automated approach for generating SIM. As also noted, mixing of S1 and RCM is likely best for targeted temporal resolution studies not large-scale products of SIM. We will continue to explore developing these high temporal resolution products that would involve mixing multiconstellation C-band SAR datasets in a future product while also operating within the bounds of available computing infrastructure.

**Reviewer #1**
Lines 175 – 180
What is a "consistent" product? This word has many meanings and should be well explained in the current scope. What is the advantage of having "consistent" product? What are the tradeoffs for generating a "consistent" product vs. a useful product? The following statistical analysis is needed for justifying "consistency": relation between decrease in coverage and increase in frequency and resolution. For example, how much can we reduce temporal averaging step and spatial resolution to keep 90% of the Arctic covered by sufficient number of vectors in each cell?

**Howell et al.**
The term "useful" is subjective. We have addressed this in a previous response.

**Reviewer #1**
Line 185
What is the impact of the threshold of 75 km/day? Where does it come from? Does it mean that vectors with 70 km/day are realistic? How many vectors are rejected? What is the impact of changing the threshold on overall accuracy and number of vectors?

**Howell et al.**
We feel this is already clear. Sometimes there is one or two really fast vectors (clearly wrong) detected on the images. Buoy comparison also showed that 60 km/day was the maximum detected so 75 km/day is just removing erroneous vectors.

**Reviewer #1**
Line 187
Why "median" is used for averaging? What is the statistical basis? Is it a normal distribution of ice motion within each cell? Should normalization be applied before averaging, so that mean can be calculated instead of median?

**Howell et al.**
Changed to mean.

**Reviewer #1**
Section 3.3
Line 219 and Lines 229 – 234
Why is the fixed uncertainty used for both satellites? First, it was shown (Holland et al., 2011; Komarov and Barber, 2014; Korosov and Rampal, 2017) that uncertainty varies from sensor to sensor. Second, the uncertainty is higher for lower cross-correlations.

What are numerical and statistical justifications for the selected formulation of the uncertainty score? Uncertainty is a valuable parameter used, for example, for model evaluation and for assimilation. It should be realistic and reflect the actual spread of drift vectors within a cell. Authors possess vast observations of drift vectors and their RMSE within each cell. And RMSE

within a cell is actually a measure of uncertainty. A statistical analysis of the relationship between the proposed formulation of uncertainty and the actual RMSE should be performed. Such analysis should clearly show impact of each component of the score: c, tau, n.

**Howell et al.**
This is a reasonable point. The spread of the vectors within a given cell is not a good indicator of uncertainty, as for example we can get just a few similar vectors in summer (leading to an unrealistically small uncertainty). So, in our uncertainty formulation we do not consider the spread of the vectors, but we provide additional information on the average correlation coefficient, number of vectors, and image coverage for a given grid cell. We have addressed this at length in a previous response.

**Reviewer #1**
Section 4.
Table 2.
The purpose of the table is not clear without some extra information. For example, it can help presenting the impact of changed resolution and frequency. The following columns should be present in the table to make it useful:
• Region
• Area
• Number of vectors per week (or per day)
• Number of pixels in the final product containing valid vectors at the following combinations:
o 7 days, 25 km
o 3 days, 12.5 km
o 1 day, 6.25 km

It will hopefully show that even at the highest resolution the number of valid pixels is not dramatically small.

**Howell et al.**
We agree that this Table is not useful and it has been removed.

**Reviewer #1**
Figure 9.
• The vectors are not visible neither in the digital, not in the printed versions of the manuscript. Fewer vectors per inch should be shown and the figures should be rasterized with much higher resolution (at least 300 dpi).
• An example of ice drift in summer and in a shoulder-seasons should be presented.
These seasons are especially challenging for ice motion retrieval from SAR.
• A line showing ice edge should be plotted to compare with the extent of the ice drift product.
Figure 10.
• Vectors are not visible here either.
• What is the source of patchiness in the drift map? Is it the discrete color scale that enhances gradients? Or is it because various patches were obtained from different image pairs and

different sensors? Or is it natural? An explanation and a proof are needed, for example as an extra map showing source of data for each vector by color.

**Howell et al.**
The original Figures were at 500 dpi. It must be the conversion software. Nevertheless, we have revised the Figures to make them easier to display. The vectors are already clipped to the ice edge as determined by NIC ice charts so there is no need to show it. Patchiness is a result of SIM heterogeneity that is typically not well-represented (almost smoothed) with PM observations

**Reviewer #1**
Section 4.2
Line 326
Due to the ambiguous nature of the alpha score in the uncertainty formulation it is impossible to associate high and low values of sigma_SIM with seasonal variations of sea ice physical characteristics. It could also be due to large range of cross-correlation coefficients, fewer image pairs, uneven distribution of images within a week. Since explicit interpretation of sigma_SIM is impossible the usefulness of the baseline values is not apparent either. As suggested above, the calculated uncertainty should be first related to the observed RMSE of vectors before its values can be interpreted.

Figure 14.
The maps with uncertainty look very heterogeneous and for the reasons mentioned above represent rather availability of data than actual uncertainty of the ice drift product. For example, one could expect overall higher uncertainty in summer and shoulder seasons (b and c) than in winter, but the range of values is equal to winter months. The uncertainty formulation should be revised to better reflect the actual spread of drift vectors within a cell.

**Howell et al.**
These are reasonable points. We have addressed uncertainty in a previous response.

**Reviewer #1**
Section 4.3
Line 356, 357
What is the purpose of averaging ice drift over a region prior to inter-comparison with OSISAF and NSIDC? Thousands of vectors are already averaged in a grid cell, so that spatio-temporal resolution of the tested and the reference products is matching. The intercomparison should therefore be performed on a cell-by-cell basis to better reflect the properties of the product on the resolution as close to the nominal resolution as possible.

**Howell et al.**
This is a good point and we have revised and provided a grid-cell to grid-cell comparison as follows:
     To facilitate a representative 1-to-1 grid cell comparison between S1+RCM SIM and both the NSIDC and OSI SAF SIM products, the spatial and temporal resolution of the S1+RCM were matched with the NSIDC and OSI SAF SIM products from March to December 2020. For OSI SAF, S1+RCM was generated with a 2-day at 62.5 km and for NSIDC, S1+RCM was generated

with 7-day temporal resolution and 25 km spatial resolution. For each product's temporal resolution (i.e. 7-day for NSIDC and 2-day for OSI-SAF), all the S1+RCM SIM vectors within each products grid cells (i.e. 25 km for NSIDC and 62.5 km for OSI SAF) were averaged. This resulted in 455,905 grid cells for the S1+RCM and NSIDC comparison and 376,386 grid cells for the S1+RCM and OSI-SAF comparison. More samples were available from NSIDC because of its higher spatial resolution.

Scatterplots of the u and v vectors components of SIM for S1+RCM versus NSIDC and OSI SAF are shown in Fig. 16 and 17, respectively. Both existing SIM products are in good agreement with S1+RCM with correlation coefficients for u and v of 0.75 and 0.78, respectively for the NSIDC and 0.84 and 0.85, respectively for OSI-SAF providing confidence in the SAR coverage for the 7-day and 3-day S1+RCM products. The RMSE is higher for the NSIDC (u=4.6 km/day and v=4.7 km/day) compared to OSI-SAF (u=3.9 km/day and v=3.9 km/day) and we note the better agreement between S1+RCM and OSI-SAF is likely because the temporal resolution more closely matches the average overlap between SAR images (i.e., ~2 days). However, the overall larger speed associated with S1+RCM is most likely the result of higher spatial resolution compared to lower resolution satellite data used in NSIDC and OSI SAF as faster speeds are more difficult to track at lower spatial resolution because of temporal decorrelation. Kwok et al. (1998) also noted this problem when comparing SIM from passive microwave with SAR and found it also applies to regions of low ice concentration. Figs. 16 and 17 also illustrate that users of either the NSIDC or OSI SAF SIM products are underestimating SIM.

[Figure]

**Figure 16.** Scatterplots of S1+RCM sea ice motion versus National Snow and Ice Data Center (NSIDC) SIM for a) u and b) v vector components.  Also shown is the number of samples (n), Pearson's correlation coefficient (R), root-mean square error (RMSE), and the mean difference (MD).

[Figure]

**Figure 17.** Scatterplots of S1+RCM sea ice motion versus Ocean and Sea Ice-Satellite Application Facility (OSI-SAF) SIM for a) u and b) v vector components. Also shown is the number of samples (n), Pearson's correlation coefficient (R), root-mean square error (RMSE), and the mean difference (MD).

**Reviewer #1**
Figure 16.
In addition to drift speed, comparison of drift direction should also be performed to form a complete picture of difference between the three products. It would also be desirable to plot a map of difference averaged, for example, over seasons.

**Howell et al.**
Since we compared both u and v components of ice motion, drift direction is taken into account in our comparison.

**Reviewer #1**
Technical corrections
L20: OSI-SAF
L54: The word "perhaps" seems to need commas around
L54: "… combining of SAR imagery …" can be misinterpreted by a reader with a meaning "Images from S1 were combined with images from RCN to provide SIM vectors". Rephrasing of the sentence is needed which clearly states that drift vectors from S1 and RCN were combined, and not imagery.
L141: What is "vertical scalability"? Some brief explanations (+ a reference) are needed.
L281: "3-day temporal resolution" contradicts the caption on Figure 12: "sea ice motion on August 12 – 18, 2020". Which is correct?
L312: The sentence "For all cases, …" needs to be rephrased. Probably: "For all cases, low sigma_SIM values are typically found in the centra Arctic with gradual increase outwards…"

L324: "significant portion of marginal ice zone

**Howell et al.**
Changed accordingly.

**Reviewer #2**
This manuscript describes a new processing setup for monitoring sea-ice motion at the pan-Arctic scale by taking advantage of satellite imagery from five C-band SAR satellite missions (Copernicus Sentinel-1 A&B, and the three missions from RCM). A first batch (10 months) of S1+RCM sea-ice motion data is prepared and evaluated qualitatively at pan-Arctic scales and regionally in the Canadian Arctic Archipelago. A comparison to two existing large-scale sea-ice motion dataset (from NSIDC and OSI SAF) reveals that the new S1+RCM data generally retrieves faster drift regimes, as well as more vectors in regions with intermediate concentrations and close to land. The paper provides a description of the processing setup, and conveys well the main message that the recent availability of operational SAR missions opens a new era for large-scale sea-ice motion mapping. The paper is convincing and can be published with some more work.

As I see it, two weak points of the manuscript at this stage are 1) the lack of dedicated quantitative validation of the new S1+RCM drift vectors, namely against trajectories from on-ice drifters, 2) the lack of a stand-alone Discussions section where the choices and assumptions made in the new processing setup and its uncertainties are justified and discussed.

**Howell et al**
We thank the reviewer for constructive comments that only serve to improve the quality of the manuscript and associated datasets.

*Summary of the Major Changes:*
1. We recast the manuscript to describe the Environment and Climate Change Canada Automated Sea Ice Tracking System (ECCC-ASITS) which is to provide routine SIM products from S1+RCM for operational needs at ECCC, the broader scientific community, and maritime stakeholders. The focus is on the latter. Accordingly, our choices, assumptions, and uncertainties are better justified throughout the manuscript.
2. Generated a new pan-Arctic 6.25 km 3-day (rolling) S1+RCM SIM product and all datasets have been updated until October 31, 2021.
3. Added validation section that compares vector displacement from S1 and RCM to buoys from the IABP.
4. Refined the uncertainty of the S1+RCM SIM products based on the buoy analysis and the time separation of the image pairs for dry and wet ice conditions.
5. Provided a 1-to-1 grid cell comparison of NSIDC and OSI-SAF SIM products to S1+RCM

**Reviewer #2 Comments**
Major comments:
Validation against buoys:
The paper would be much stronger with a dedicated validation against buoy data at the pan-Arctic scale. Validation against buoy data is the de-facto standard for documenting the accuracy of sea-ice drift datasets (e.g. OSI SAF, NSIDC, Kwok 1998, etc…). In your case it would be particularly useful because validation of RCM SIM vectors (and thus geo-location, resolution, speckle) have never been assessed. You could also check the assumptions built into your uncertainty model (e.g. the scaling of the uncertainty on velocities by Delta_t, see discussion

below). I strongly suggest that a dedicated validation against buoy data is conducted and reported here, but leave it to the Editor to decide if this major revision is required or not.

**Howell et al.,**
Although comparison of the algorithm against buoy data has been done before, we agree it is important to re-assess with new sensors (although still at the same frequency).  We attached the resulting comparison for against winter (April) and summer (August) buoys for 2-years for the reviewer's reference:

[Figure]

**Figure 14.** Comparison between ice motion vectors derived by the Komarov and Barber (2014) automated sea ice tracking algorithm from S1 and RCM SAR images and buoy data.

Based on the comparison we are develop to uncertainty estimates for dry and wet conditions as follows:

In order to estimate the SIM uncertainty from the ECCC's automated SIM tracking algorithm, we compared SIM displacement vectors from S1 and RCM to buoy positions from IABP during winter and summer time periods.  For all S1 and RCM displacement vectors (derived from image pairs), the closest buoy trajectory was co-located to the start of each displacement vector position. The distance between the starting point of a given SAR ice motion tracking vector and the starting point of the corresponding buoy trajectory did not exceed 3 km. Fig 13. summarizes the results for dry winter conditions (April 2020 and 2021) and during the melt season (August 2020 and 2021). The ECCC automated SIM tracking algorithm performs very well during winter conditions with a root mean square error (RMSE) of 2.78 km and a mean difference (MD; bias) of 0.40 km.  The RMSE is higher than the value reported by Komarov and Barber (2014) likely because more image pairs over a larger geographical area were used in this comparison as well as the spatial resolution was lower. Performance decreases during the summer with a lower number of vectors detected and an RMSE of 3.43 km.

Taking into considering the difference between the winter and the summer we assign two uncertainties to the S1+RCM SIM products for dry and wet conditions as follows. Consider a grid cell containing a set of $N$ sea ice velocity vectors $\vec{V}_i$, where $i = 1, 2, ..., N$. Each vector has the

following uncertainty associated with the SIM tracking algorithm deriving the ice motion vector from two consecutive images:

$$\Delta V_i = \frac{s_0}{\Delta t_i},$$
(1)

where, $\Delta t_i$ is the time interval (in days) separating two SAR images used to derive the considered ice velocity vector $\vec{V_i}$. $s_o$ is the uncertainty in sea ice displacement (not speed) for dry ice conditions (2.78 km) or wet ice conditions (3.43 km). Note that $s_o$ must be divided by $\Delta t_i$ to come up with the ice velocity uncertainty. The average uncertainty for dry ($s_o = 2.78$ km) and wet ($s_o = 3.43$ km) ice conditions in each grid cell (N) is then determined using the following equation:

$$\sigma_{SIM} = \frac{1}{N}\sum_{i=1}^{N}\frac{s_0}{\Delta t_i}$$
(2)

[Figure]

**Figure 15.** Spatial distribution of (a) dry and (b) wet S1+RCM SIM uncertainty for August 5-11, 2020

**Reviewer #2**
Sea-ice motion technique:
Section 3.2 is missing some details to fully characterize the processing. Some of the missing elements are:
* how old are the scenes allowed to be before they are not taken in the stack of scenes?
* In Fig. 5: we see that IMG1xIMG2, IMG2xIMG3, IMG3xIMG4, etc… are processed for SIM, but what about IMG1xIMG3, IMG1xIMG4, etc…? Considering these overlaps would dramatically increase the number of retrieved vectors and the sampling in the temporal domain. Please indicate if these additional overlaps are processed for SIM and, if not, add a discussion/justification why they were not considered (e.g. in a Discussions section).

**Howell et al.**
This is important to note. We have added the following to section 3.2:

It is important to note there is currently no "staleness" limit for images in a given sector. There are occasionally instances when long stretches of time (e.g., 7 days) occur between images pairs but this is mostly confined to the edge-sectors of the grid. Unfortunately, the computational capacity to take on the additional processing load of using the same image in multiple pair combinations is not currently available in the infrastructure being used.

**Reviewer #2**
* In Fig. 3: it is clear and well justified that S1 and RCM scenes are processed on their own (before the merging step). Are SIM vectors processed within the S1 and RCM missions? E.g. S1a with S1b, RCMa with RCMc, etc… Please add this information.

**Howell et al.**
The following statement has been added to:
S1A and S1B are freely mixed in the Sentinel processing chain as well as RCM1, RCM2 and RCM3 are mixed in the RCM processing chain.

**Reviewer #2**
* Fig 6 a) gives the impression that S1 has a complete coverage of the dark blue region at least once on every week. Is it really the case, or are there weeks were S1 leaves some holes in the weekly coverage? Could these [0-1] average density be in a different color to better appreciate the weekly coverage? Same for b).

**Howell et al.**
This is a very good suggestion and yes, it is very close that there is almost complete coverage by S1 once a week. We changed the legend of Figure 6 (now Figure 3) to a quantile to better illustrate the coverage.

[Figure]

**Figure 3.** Image density per week for a) S1, b) RCM, and c) S1+RCM based on images from March 2020 to October 2021.

**Reviewer #2**
* L184: what is the justification for the cap at minimum 12 hours?
* starting L183: it is not immediately clear that you average the velocity vectors instead of the displacement vectors. Please clarify.

**Howell et al.**
i) We justified the 12 hour cap as follows:

We selected a 12 hrs cut-off because below 12 hrs the SIM resulted in less representative (usually higher speeds) with respect to the averaged product value (over 3 or 7 days). This was the primary observation from previous studies constructing a very high temporal resolution time series (e.g. Howell and Brady, 2019; Moore et al., 2021a). This is likely related to uncertainty, as vectors with lower time separation are more uncertain especially at sub-daily time intervals to be used for 3 or 7 days average SIM products

ii) We are averaging the velocity vectors and not the absolute speeds. This is now clarified in the methods.

**Reviewer #2**
Minor comments:
L49: The dataset based on passive microwave indeed have coarse resolution, that rather are in the range (50 – 100 km) than (12-25 km) as stated here. The OSI SAF is ~60km like the data from IFREMER/CERSAT, Kwok's is ~100km. NSIDC's 25km grid results from oversampling (see e.g. Table 2 of the NSIDC V004 User Guide).

**Howell et al.**
Changed.

**Reviewer #2**
L75: Did you use the multi-sensor OSI SAF product (multi-oi) or the single-sensor products (from AMSR2, SSMIS, etc…) Please provide this information.

**Howell et al.**
We used the multi-sensor low resolution 62.5 km gridded products (OSI-405). [https://osisaf-hl.met.no/osi-405-c-desc](https://osisaf-hl.met.no/osi-405-c-desc)

**Reviewer #2**
Fig 14: the labels and legends are hardly readable. Please enlarge the text.

**Howell et al.**
Figure 14 has been removed.

**Reviewer #2**
Conclusions: with "swath-to-swath" approach, SIM from passive microwave now achieves sub-daily temporal resolution (Lavergne et al. 2021). This will be extremely difficult to reach consistently and pan-Arctic from SAR constellations alone. Maybe the complementary of SIM estimation from SAR and "swath-to-swath" PMW would deserve a mention in the Conclusions.

**Howell et al.**
Good point. We have added this to the Conclusions as follows:

While groups like the Polar Space Task Group aim to improve or refine SAR coverage across the pan-Arctic over the annual cycle it is unlikely a purely SAR derived SIM product will be able to achieve daily or sub-daily consistently across the pan-Arctic. This has recently been achieved with passive microwave observations using a swath-to-swath approach (Lavergne et al., 2021). Therefore, it could be worth exploring the complimentary of SIM provided from passive microwave "swath-to-swath" and SIM generated from SAR.

**Reviewer #2**
Editorials:
L15: delete "able to be"
L18-19: OSI SAF, without "-" (in long form and acronym).
L49: "trade-off with respect to" → "drawback of" or "limitation of".
L73: replace "2020" with "this period".
L86: here and later in the section: "coarser spatial resolution levels". Consider changing "levels" with "images" for clarity.
L90. Lowest resolution → coarsest resolution
L135. "at _least_ 32,000 km2"
L250. Based _on_ the weekly image….

**Howell et al.**
All changed.

---

## Referee Report (RR1)

The manuscript and, more importantly, the dataset have been significantly improved. The authors generated a new sea ice motion product at higher temporal and spatial resolution and extended the temporal coverage until October 2021. The manuscript now clearly presents the potential users of the product and the scope of its application. The decision for selecting spatial and temporal resolutions and not mixing S1 and RCN imagery is now clearly justified. The high-quality maps of SIM at lower and higher resolution show the advantages of the proposed new dataset. The methodology for uncertainty computation was simplified and made more robust and versatile. Accuracy of the S1+RCM SIM product is computed by validating against IABP buoys and pixel-by-pixel comparison with NSIDC and OSI-SAF products.

The manuscript consistently documents the ECCC automated sea ice tracking system and the generated sea ice motion product and as such can be recommended for publication after a minor revision.

**Minor comments**

Table 2 seems to provide too many numbers that are not very relevant for most readers. I would suggest, first, to replace the absolute number of grid cells with a number relative to ice covered cells, or at least to provide the average number of ice-covered cells for the low- and high-resolution products. Second, as seasonality seem to repeat in 2020 and 2021, the number of cells can be averaged for each month, or even each season. Thus, the table can contain just 8 numbers: relative coverage by the low- and high- resolution product in winter, spring, summer, and autumn.

**Technical comments**

L107. ECCC-ASITS is spelled as ECCC-ASTIS in several places.

L167: between image pairs

L221: What is the average number of grid cells for the high-resolution product? What is the average number of grid cells covered by sea ice?

L222:  is shown

L223: "especially, during" – the comma doesn't seem to be needed here

L275: "because of its narrow"

L278: "increase"

---

## Author Response (AR2)

Dear Dr. Brucker,

We have responded to all the comments/suggestions provided by Reviewer #1 and Reviewer #3. We implemented almost all the suggestions from both Reviewers. In fact, Reviewer #3 had some excellent points that in our opinion significantly improved the manuscript.

Summary of changes:
- Removed Table 2
- Added text and new Figure (Fig. 12) concerning image sampling and high resolution time series construction in Section 4
- Added more text and additional references concerning uncertainty in Section 5

Steve, Alex, and Mike

**Reviewer #1**
The manuscript and, more importantly, the dataset have been significantly improved. The authors generated a new sea ice motion product at higher temporal and spatial resolution and extended the temporal coverage until October 2021. The manuscript now clearly presents the potential users of the product and the scope of its application. The decision for selecting spatial and temporal resolutions and not mixing S1 and RCN imagery is now clearly justified. The high-quality maps of SIM at lower and higher resolution show the advantages of the proposed new dataset. The methodology for uncertainty computation was simplified and made more robust and versatile. Accuracy of the S1+RCM SIM product is computed by validating against IABP buoys and pixel-by-pixel comparison with NSIDC and OSI-SAF products. The manuscript consistently documents the ECCC automated sea ice tracking system and the generated sea ice motion product and as such can be recommended for publication after a minor revision.

**Howell et al.**
We thank the Reviewer for their comments, which have improved the manuscript considerably.

**Reviewer #1**
Minor comments
Table 2 seems to provide too many numbers that are not very relevant for most readers. I would suggest, first, to replace the absolute number of grid cells with a number relative to ice covered cells, or at least to provide the average number of ice-covered cells for the lowand high resolution products. Second, as seasonality seem to repeat in 2020 and 2021, the number of cells can be averaged for each month, or even each season. Thus, the table can contain just 8 numbers: relative coverage by the low- and high- resolution product in winter, spring, summer, and autumn.

**Howell et al.**
We just decided to remove Table 2. Reviewer #3 pointed out they did not find it useful either.

**Reviewer #1**
Technical comments

L107. ECCC-ASITS is spelled as ECCC-ASTIS in several places.
L167: between images pairs
L221: What is the average number of grid cells for the high-resolution product? What is the average number of grid cells covered by sea ice?
L222: are is shown
L223: "especially, during" – the comma doesn't seem to be needed here
L275: "because of its narrow"
L278: "increased

**Howell et al.**
Re: average number of grid cells → This information is contained in the each product.
All the other comments have been addressed.

**Reviewer #3**
First, I apologize to the authors and editors for being so tardy in my review.

Review, Howell et al, Generating large-scale sea ice motion from Sentinel-1 and the RADARSAT Constellation Mission using the Environment and Climate Change Canada automated sea ice tracking system, Cryosphere tc-2021-223

This is a good summary of the SAR-derived ice motion product using two different C-band SAR sensors on 5 different platforms. I reviewed the revision made after comments provided by Reviewers 1 and 2 but I did not read the reviewers comments or response until after I read the revision. I certainly agree with many of these reviewers' comments and revisions, specifically the inclusion of the 6.25 km product and the buoy comparisons. I also appreciate the revisions made by the authors, to emphasize the primary purpose of these two products which is for marine stakeholders. I believe it is very important to have papers such as this on describing approaches and algorithms for new products, especially those derived from satellite, particularly in journals where the papers will be read by the sea ice community in this case. I assume the editors of TC believe this paper and such related papers on products/algorithms are appropriate, once approved for submission of course through the review process.

I follow now with my review. In general, I found the paper well-written and worthy of publication after my relative minor suggestions are evaluated and commented on. I appreciate the scale of this effort to provide SAR ice motion products. I believe the paper could use some more discussion on some figures which I will mention. I am a little surprised that the buoy comparisons seem quite large especially on the basis of the selected grid size used for 6.25 km product and suggest that more description be included on the multiple sources of the errors (Section 5). The offsets are actually pretty large compared to previous publications using coarser resolution SAR imagery. Also, in comparing the SAR 7-day product with PM results (Section 6), it's understandable that the SAR product is better. I think this section could benefit from more discussion especially on why stakeholders might be more interested in using the SAR product vs. the PM, aside from the improved accuracy.

**Howell et al.**

We thank this reviewer for their comments which have further improved the manuscript. We have implemented almost all of their suggestions.

**Reviewer #3**
Detailed comments
1. Lines 61-62. Please rephrase this sentence to include the thought that the SAR data limitations prior to S1.

**Howell et al.**
Good suggestion.  Rephrased as follows:
Prior to S1 there was a lack of widely available SAR imagery for pan-Arctic SIM generation and as a result, this is perhaps the first time such an extensive processing of SAR imagery at the pan-Arctic scale has been undertaken to generate SIM.

**Reviewer #3**
2. Line 65. Please add a sentence that mentions the rationale for near-real time SIM processing and derived products. The processing scenario would be more seamless without the need for near-real time processing. Helps explain need for Figure 7 discussion too.

**Howell et al.**
Good suggestion.  Rephrased as follows:
Here, we focus primarily on the latter applications by first describing the ECCC-ASITS workflow that produces SIM from S1 and RCM SAR imagery (hereafter, S1+RCM) in close to near-real time and combines the output into S1+RCM SIM products. A close to near-real time workflow is required given the considerable amount of incoming SAR imagery together with the computational time to generate SIM.

**Reviewer #3**
3. Table 1. I suggest adding the PM sea ice products to this table or making an equivalent one, including pixel size and grid size, temporal spacing (not image count).

**Howell et al.**
Considering the PM SIM products are relatively simple to describe, dedicating another Table to them is not warranted.  This information is easily communicated to the reader (simply) in text form. We did note that the pixel size for the NSIDC product was missing in the text, which is now added.

**Reviewer #3**
4. Sampling. From Figure 3, it seems pretty clear that the RCM mapping strategy in general is different from S1 for the Arctic. RCM has a more spatially/temporally distributed consistent sampling approach for the entire Arctic, while S1 has more intensive coverage in certain areas. Each mission is set up to meet its science/stakeholder requirements. Given the overlapping coverage shown in Figure 3C, which shows a major central portion of the Arctic with over 9 images per week, I am surprised at the sparse results shown later in Figure 10, since 5 sensors are being utilized. The enlargements shown in a-d suggest more extensive results than the large figure but I don't find these that useful. I wonder if an additional large figure or two showing

sequential results, say from March 15-17, then March 18-20, would be beneficial to show. Such additions might highlight some of the issues with uneven spatial sampling that both acquisition plans may have, perhaps, and would also indicate how a regional time series could be developed for further study.

**Howell et al.**
The RCM image sampling strategy is based on our ordering from ECCC. That is, we aim to cover the majority of the pan-Arctic domain every 3-days specifically for SIM generation. However, RCM coverage is not systematic so consistent coverage across the entire Arctic is not always achievable week to week. S1 on the other hand is systematic and provides consistent coverage. Figure 3 is an average of the entire record (March 2020 to October 2021) so there is some variability that is masked at 3-days. For example, the gap in the Laptev Sea, Hudson Bay, and Beaufort Sea are often present at 3-days but not 7-days. The other point to remember is just because imagery is available, does not mean automatic SIM detection will be successful uniformly across the image. To this end, we understand the Reviewer's point and suggestion as this ultimately highlights the difficultly in generating spatially consistent 3-day products from these image sources. We have added another Figure (Fig. 12) as suggest and added some text related to sampling. We have decided to keep the insets because we feel they highlight the spatial resolution of the actual product, not reduced for presentation clarity.

New and revised Section 4 text:
        The spatial distribution of 6.25 km 3-day pan-Arctic S1+RCM SIM for selected periods during the winter and summer are is shown in Fig. 10 and Fig. 11, respectively. The insets of both Fig. 10 and Fig. 11 illustrate the level of SIM spatial detail captured at 6.25 km. More spatial gaps across the pan-Arctic using higher spatiotemporal resolution especially during the summer months. These problems relate to the challenge of constructing a complete picture of pan-Arctic SIM every 3-days using available SAR imagery because of their different acquisition scenarios. Specifically, RCM acquisitions are more spatiotemporally distributed across the Arctic whereas, S1 are more intensive in certain regions (Fig. 3) and as a result, regions can be missed on certain days. This uneven spatial imaging problem is illustrated in Fig. 12 where it is apparent SIM is captured in the Beaufort Sea, Chukchi Sea and Hudson Bay from March 12-14, 2021 but absent from March 14-16, 2021. Also, just because SAR image pairs are available over a region it does not imply automatic SIM detection will be successful, especially during the summer months. Despite this, there are still many regions across the Arctic where high spatial and temporal SIM can be resolved using S1+RCM but the aforementioned problems need to be taken into consideration with respect to regional time series development.

[Figure]

**Figure 12. The spatial distribution of 6.25 km 3-day S1+RCM sea ice motion on a) March 12-14, 2020 and b) March 14-16, 2020. Note that the white areas in the figure indicate either zero ice motion for the landfast ice or no ice motion information extracted (because of no SAR data, no ice, or no stable ice features).**

**Reviewer #3**

Continuing on about sampling, as Reviewer 1 mentioned, and has mentioned in Lines 193-194 and paragraph starting with line 195, I too would be interested in hearing whether they tried mixing RCM and S1image pairs to derive a motion product and what the results were.

**Howell et al.**

Stay tuned. This is a work in progress given the recent problems with S1B. Since late December 2021, S1B is not available and SIM is "gapy" across the Arctic. Our strategy is to keep the near-real time processing chain going but run scenarios that mix S1B and RCM to back process SIM from January 2022 onward to see if we can improve coverage. If successful, this "mixing routine" will be used to generate a more research based SIM product. If we are able to increase computational capabilities and receive the imagery from the ground stations more consistently, we will work this into the processing chain. This will be a considerable amount of work.

**Reviewer #3**

Also, regarding use of stacks, perhaps I missed this but for high coverage areas, say within the central Arctic, how do they handle more than one image pair within a 3-day period- are multiple pairs averaged together to make a single 3-day product?

**Howell et al.**
Yes.

**Reviewer #3**
5. Section 4

I assume 'resolvable' means grid cells with derived vectors that passed quality check. Please clarify.

**Howell et al.**
Passed the quality check and detected by the algorithm. This is pretty clear.

**Reviewer #3**
The availability of 5 sensors collecting Arctic sea ice imagery has enabled the most complete picture of Pan-Arctic SAR-derived ice motion. Please clarify sentence starting on Line 214. Its long been desired.

**Howell et al.**
Revised a follows:
The 7-day 25 km spatiotemporal scale is able to provide the most complete picture of SIM across the pan-Arctic from SAR imagery alone, this has long been desired.

**Reviewer #3**
Neither Table 2 or the text really addresses the point about the most complete pan-Arctic SAR-derived maps, at least in terms of numbers.

**Howell et al.**
We decided to remove Table 2.
**Reviewer #3**
6. Section 5
In the first paragraph, the authors should include the reference Lindsay and Stern, 2003 on RGPS offsets, where a mean displacement of 323 m was found for Radarsat-1

Paragraph starting line 309. As mentioned errors of 2.78 km seem quite large to me, basically since its about half of 6.25 km grid cell for winter, also in comparison to the above Lindsay/Stern results. I strongly suggest the authors add a paragraph that describes in more detail the sources of error- buoy, interpolated time to SAR image, spacecraft and SAR image location accuracy, identifying matchups between image pairs and so forth. This would at least enable readers and authors to point to major error sources which may be improved upon.

**Howell et al.**
This is a good suggestion. Briefly, we considered all buoys north of 40N and not just the Central Arctic like Lindsay and Stern (2003). As a result, considerably more ice regimes where taken into consideration during the winter and summer. Our RMSE is half of Wilson et al. (2001) who looked at the dynamic region of Baffin Bay but the Wilson et al. (2001) tracking algorithm is not as sophisticated as ours. Indeed geolocation errors and methodological choices contribute to uncertainty and both need to be acknowledged.

We have revised this section as follows:

…"The only restrictions placed on the buoys where that they were located north of 40N and the distance between the starting point of a given SAR ice motion tracking vector and the starting point of the corresponding buoy trajectory did not exceed 3 km."…

Our RMSE is higher than the value reported by Komarov and Barber (2014) likely because the initial validation assessment of the automated tracking algorithm only used 35 sample points in the Beaufort Sea during the winter and the vectors were at a higher spatial resolution (i.e. 100 m). Our RMSE is slightly higher than reported by Lindsay and Stern (2003) who compared the RGPS SIM to buoys in the Central Arctic and reported an RMSE of ~1 km for the winter and ~2 km for the summer. However, our RMSE estimates are much lower than Wilson et al. (2001) who compared RADARSAT-1 SAR estimates of SIM to buoys in Baffin Bay and reported an RMSE of 3.8 km for the winter and 6.8 km for the summer. The differences in RMSE's can be attributed to numerous factors including the geolocation errors of the different SAR satellites, differences in the methodology for buoy comparison, and different tracking algorithms. Overall, our validation is certainly representative of large-scale SIM uncertainty because we considered a wide-range of ice conditions during both the winter and summer months.

**Reviewer #3**
Figure 15A- the date on figure and in caption are mislabelled. For Figure 15B I found it interesting that a couple of locations within the CAA showed higher velocities (yellow) than surrounding areas. This figure warrants additional discussion.

**Howell et al.**
The date is correct and is meant to be the same in both panels. Panel (a) illustrates what the uncertainty would look like for dry conditions and the right (b) for wet conditions. SIM is very heterogeneous within the CAA so this is to be expected. However, discussion of this Figure would be related to regional SIM in the CAA and that is not the topic of this paper.

**Reviewer #3**
7. Section 6
I too am a little puzzled by comparing SAR with PM, including down-sampling the SAR results. But the PM products do at least have similar spatial scales to the weekly product. I am uncertain as to the meaning of the colors in Fig 16 and 17. Both appear to be heat maps. Both figures warrant additional discussion beyond lines 371-2.

**Howell et al.**
We wanted to do this comparison because the PM products are so widely utilized. This also gives confidence to all products should they achieve similar results. More importantly, we wanted to quantitatively demonstrate the speed bias inherent with PM sensors versus SAR. Admittedly, this was not well articulated in our initial text and some was incorrectly placed at the end of Section 5. The Figures are heat maps and this is now described in the Figure Captions. We do not feel a detailed analysis of this topic is warrant as it has already been done by Kwok et al. (1998).

We have added an introduction to Section 6 as follows:

Given the difficultly in quantifying SAR image pair coverage on S1+RCM SIM uncertainty, we now compare S1+RCM SIM to the NSIDC and OSI SAF SIM products that are widely utilized by the sea ice community. Such a comparison provides additional quantitative confidence metrics to assess the quality of the S1+RCM SIM estimates.

**Reviewer #3**
Details:
Line 33- Torres is misspelled
**Howell et al.**
Changed.

**Reviewer #3**
Line 41- Kwok 2015 paper is not listed in the references
**Howell et al.**
Added.

**Reviewer #3**
Lines 122-123 not a sentence
**Howell et al.**
Finally, different orbit characteristics of the satellites contribute to timing differences between when the images are acquired compared to when they are received by ECCC.

**Reviewer #3**
Line 122. How much computer time does it take to analyze 160 images per day?
**Howell et al.**
The system is constantly running ever hour.  We don't have those specific metrics.

**Reviewer #3**
Lines 153-155 shouldn't the overlap be greater than 30%, for retention?
**Howell et al.**
Yes. Changed.

**Reviewer #3**
Line 156- you might mention that 32,000 km/2 is about 178 km2 or a little less than 1/2 a frame.

**Howell et al.**
We are OK with 32,000 km^2. The frame can be variable depending on the sensor. Further, some imagery we receive via the Canadian Ice Service is not a complete frame!

**Reviewer #3**
Line 191 – spatially complete
**Howell et al.**
Changed.

**Reviewer #3**
Line 198-199 – insert: …less representative results, often due to higher wind speeds.

**Howell et al.**

Done.